# TGFβ2-induced formation of lipid droplets supports acidosis-driven EMT and the metastatic spreading of cancer cells

Cyril Corbet[1,4]*, Estelle Bastien[1,4], Joao Pedro Santiago de Jesus[1,4], Emeline Dierge[1], Ruben Martherus[1], Catherine Vander Linden[1], Bastien Doix[1], Charline Degavre[1], Céline Guilbaud[1], Laurenne Petit[1], Carine Michiels[2], Chantal Dessy[1], Yvan Larondelle [3] & Olivier Feron [1]*

Acidosis, a common characteristic of the tumor microenvironment, is associated with alterations in metabolic preferences of cancer cells and progression of the disease. Here we identify the TGF-β2 isoform at the interface between these observations. We document that acidic pH promotes autocrine TGF-β2 signaling, which in turn favors the formation of lipid droplets (LD) that represent energy stores readily available to support anoikis resistance and cancer cell invasiveness. We find that, in cancer cells of various origins, acidosis-induced TGF-β2 activation promotes both partial epithelial-to-mesenchymal transition (EMT) and fatty acid metabolism, the latter supporting Smad2 acetylation. We show that upon TGF-β2 stimulation, PKC-zeta-mediated translocation of CD36 facilitates the uptake of fatty acids that are either stored as triglycerides in LD through DGAT1 or oxidized to generate ATP to fulfill immediate cellular needs. We also address how, by preventing fatty acid mobilization from LD, distant metastatic spreading may be inhibited.

[1] Pole of Pharmacology and Therapeutics (FATH), Institut de Recherche Expérimentale et Clinique (IREC), UCLouvain, 57 Avenue Hippocrate B1.57.04, B-1200 Brussels, Belgium. [2] Laboratory of Cellular and Molecular Biology (URBC), Namur Research Institute for Life Sciences (NARILIS), University of Namur, 61 rue de Bruxelles, B-5000 Namur, Belgium. [3] Louvain Institute of Biomolecular Science and Technology (LIBST), UCLouvain, Croix du Sud 2/L7.05.01, B-1348 Louvain-la-Neuve, Belgium. [4] These authors contributed equally: Cyril Corbet, Estelle Bastien, Joao Pedro Santiago de Jesus. *email: cyril.corbet@uclouvain.be; olivier.feron@uclouvain.be

Acidosis, like hypoxia, is a hallmark of the tumor micro-environment (TME)[1–6]. It largely arises from the conjunction of the exacerbated metabolism of cancer cells (and cancer-associated cells) with a spatio-temporal disorganization of the tumor vasculature which limits the access to $O_2$ and prevents the rapid elimination of waste products including protons ($H^+$) and $CO_2$[7]. The most common source of $H^+$ is the one associated with lactate production from glucose metabolism[8] but hydration of $CO_2$ molecules generated upon decarboxylation of metabolic intermediates also leads to the production of $H^+$ [9,10].

We and others have recently provided evidence that tumor acidosis could in turn influence the metabolism of cancer cells[11–16]. We have for instance reported that glutamine metabolism is preferred to glucose metabolism when cancer cells are maintained at pH 6.5, thereby limiting the rate of $H^+$ generation from glycolysis[14]. We also documented that fatty acid (FA) metabolism is profoundly altered in response to ambient acidosis with FA oxidation (FAO) acting as a main source of acetyl-CoA to support TCA cycle[15]. Moreover, acidosis-mediated metabolic rewiring is closely associated with sirtuin-mediated deacetylation of hypoxia-inducible factors HIF-1α and HIF-2α as well as histones H3 and H4 in the nucleus, leading to the differential expression of several metabolism-related enzymes[14,15]. In addition, acidosis can abolish hypoxia-induced gene reprogramming, in particular by inhibiting HIF-1α protein stabilization[17,18]. Altogether, these data indicate that acidosis, on its own, may strongly influence cancer cell metabolic phenotype.

Tumor progression is well known to be driven by an acidic extracellular pH through both evasion from immunosurveillance and an increased invasive potential[1]. It is for instance documented that accumulation of $H^+$ and lactate in the TME decreases the production of cytokines by several immune cell populations[19,20] and that acid-activated proteases facilitate cancer cell migration[21–23]. By contrast, the relation between metabolic preferences under acidic pH and local/metastatic invasiveness only begins to be explored[24,25]. The link between the ability of cancer cells to generate metastases and their bioenergetics is, however, increasingly documented[26–32]. New insights on the interplay between acidosis-driven metabolic preferences and a potential invasive behavior are therefore of particular need.

Here we identified lipid droplets (LD) as a common denominator to acidosis-adapted cancer cells issued from different tissues and a critical driver of their increased invasiveness (vs. native cells maintained at neutral pH). A transcriptomic study comparing both cell populations led us to identify TGF-β2 as a transcript consistently upregulated in the different acid-adapted cancer cells. We expanded on this finding to document how lipid metabolism rewiring is associated with partial epithelial-to-mesenchymal transition (EMT). In particular, we documented how autocrine TGF-β2 signaling favors the formation of lipid droplets (LD) that further act as energy stores to support anoikis resistance, local invasiveness and distant metastatic spreading.

## Results
### Acidosis-adapted cancer cells accumulate lipid droplets. The stimulated FA metabolism reported in acidosis-adapted cancer cells[15] led us to identify intracellular vacuole-like structures detected by electron microscopy in these cells as being lipid droplets (LD; positive for BODIPY 493/503 and Oil Red O staining) (Fig. 1a–c); a consistent increase in the cellular area covered by LD was observed in pH 6.5-adapted cancer cells (named 6.5/cancer cells here below) vs. native cells maintained at pH 7.4 or 7.4/cells (Fig. 1d and Supplementary Fig. 1a); 6.5/cancer cells (obtained after 4–6 weeks of culture at pH 6.5) were

maintained at pH 6.5 for the rest of this study. LD accumulation could not be attributed to cell arrest as reported in other situations[33] since in our experimental setup, acidosis-adapted cancer cells keep proliferating (although often to a smaller extent) contrary to acutely acid-exposed cancer cells (Supplementary Fig. 1b–d). A lipidomic analysis confirmed a net increase in neutral lipids in 6.5/cancer cells (Fig. 1e and Supplementary Fig. 1e), mainly represented by a gain in saturated and mono-unsaturated FA but not cholesteryl esters (CE) (Fig. 1f, g, Supplementary Fig. 1f, g, and Supplementary Table 1); in comparison, the pools of phospholipids were not altered in the different cell populations (Supplementary Fig. 1h).

We next searched to identify actors directly involved in the accumulation of neutral lipids in acidosis-adapted cancer cells. We first examined the expression of perilipins (PLIN), a family of proteins that associate with LD surface and enzymes involved in the synthesis of neutral lipids. We found that, among major members of the PLIN family, PLIN2 mRNA and protein expression was consistently increased in each of the tested 6.5/cancer cells while other PLIN isoforms remained unaltered (Fig. 1h and Supplementary Fig. 1i–k). We further documented that in acidosis-adapted cancer cells, *PLIN2* silencing using four siRNA duplexes designed to target distinct gene sites (Dharmacon) significantly reduced LD accumulation (Fig. 1i). We then evaluated a series of pharmacological inhibitors or blocking antibodies targeting major proteins that mediate triglyceride (TG) and CE synthesis (Fig. 1j). It should be noted that in our hands, acidosis-adapted cancer cells were particularly resistant to plasmid or viral transduction and/or died during the selection procedure, further supporting the use of pharmacological inhibitors (or siRNA) instead of stable gene silencing approaches. We found that A922500, a diacylglycerol acyltransferase DGAT1 inhibitor, largely inhibited LD reformation contrary to PF-06424439, a DGAT2 inhibitor (Fig. 1k). Inhibitors of HMG-CoA reductase (simvastatin) and ACAT (avasimibe), as well as the use of lipoprotein-deficient serum, failed to influence LD formation (Supplementary Fig. 1l), in agreement with the lack of differences in the extent of CE between native and acidosis-adapted cancer cells (Fig. 1g and Supplementary Fig. 1g). The glutaminase inhibitor BPTES that we showed to block lipid synthesis in acidosis-adapted cancer cells[15] also failed to change the extent of LD in these cells (Supplementary Fig. 1m). On the contrary, we could document that LD formation was only observed in the presence of (lipid-containing) full serum but not charcoal-delipidated serum (Fig. 1l); addition of exogenous FA to the latter restored LD biogenesis (Fig. 1l and Supplementary Fig. 1n). Finally, we identified CD36 as a main entry path for exogenous FA, since the use of specific blocking antibodies (JC63.1 and FA6-152) prevented LD formation (Fig. 1m) as well as the uptake of a fluorescent palmitate analog (BODIPY-conjugated $C_{16}$) in acidosis-adapted cancer cells (Supplementary Fig. 1o). Altogether these data indicate that chronic acidosis induces LD formation in cancer cells, with CD36 and DGAT1 as key players to mediate LD biogenesis through the uptake of exogenous FA and triglyceride synthesis, respectively.

### Lipolysis supports cancer cell survival and invasiveness. We then investigated the role of LD in acidosis-adapted cancer cells. First, since acidosis-adapted cancer cells take up large amounts of exogenous FA, we reasoned that storage into LD could prevent lipotoxicity. To examine this hypothesis, cells were treated with oleic acid (OA), a potent inducer of TG synthesis that becomes toxic for cells incapable of handling excess neutral lipids[34]. Consistent with a reduced capacity of FA storage into LD, OA exposure preferentially led to growth inhibition in PLIN2-silenced acidosis-adapted cells (Fig. 2a and Supplementary Fig. 2a). OA also induced ER stress as detected by BiP expression,

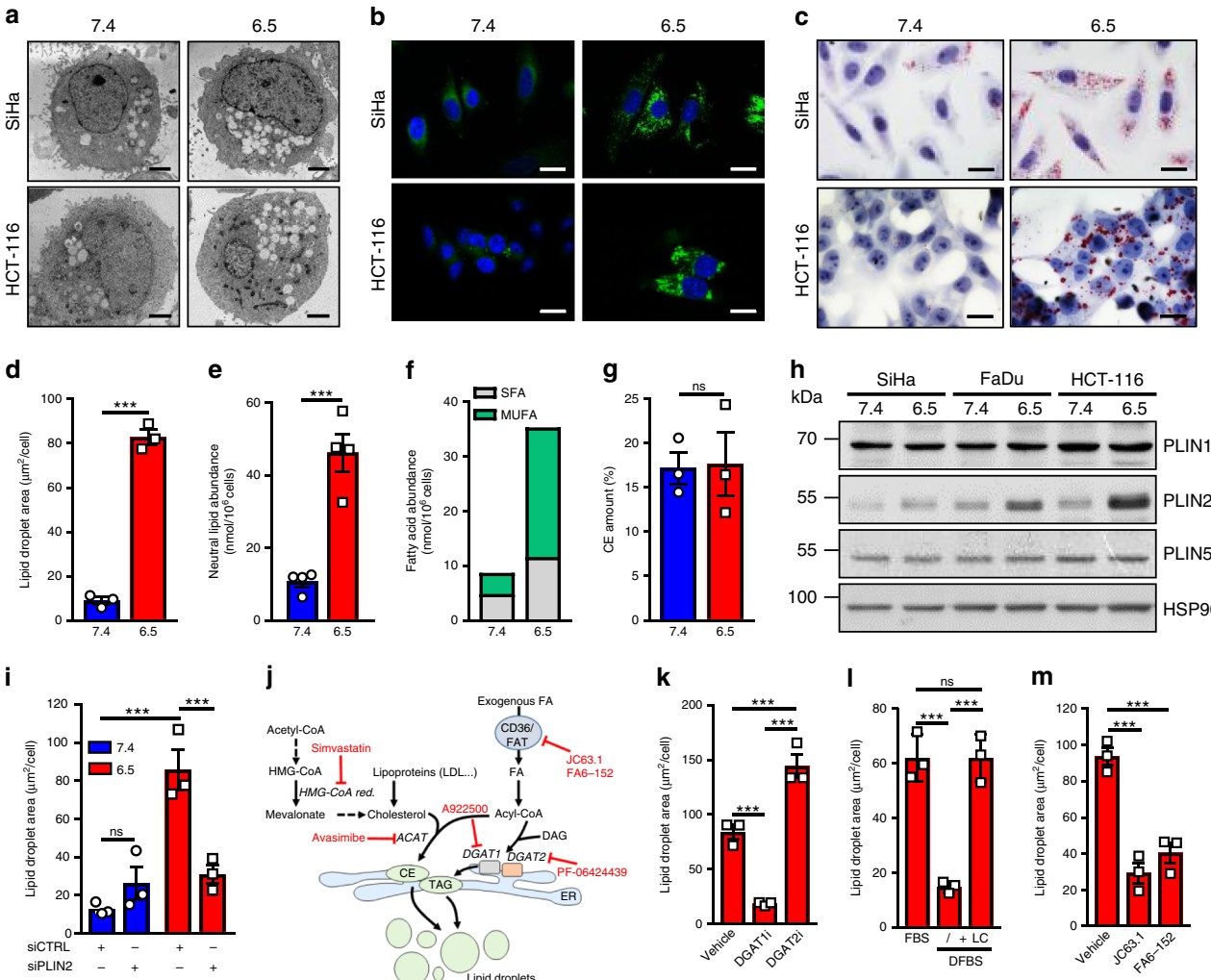

**Fig. 1 Acidosis-adapted cancer cells exhibit accumulation of lipid droplets. a** Representative electron microscopy pictures depicting lipid droplet (LD) accumulation in acidosis-adapted SiHa and HCT-116 cancer cells. Scale bar: 2 μm. **b**, **c** Representative pictures of BODIPY 493/503- (**b**) and ORO-stained (**c**) LD in the same native and acidosis-adapted cancer cells. Scale bar: 20 μm. **d** Quantification of ORO-stained LD area in native and acidosis-adapted SiHa cells. **e**–**g** Abundance of the total pool of neutral lipids (**e**), SFA and MUFA in the neutral lipid fraction (**f**), and cholesteryl esters (**g**) in native and acidosis-adapted SiHa cells. **h** Representative immunoblotting for perilipins 1, 2, and 5 in native and acidosis-adapted cancer cells. **i** LD content in native and acidosis-adapted SiHa following transfection of PLIN2-targeting (or control) siRNA for 72 h. **j** Scheme depicting the pathways leading to neutral lipid accumulation and lipid droplet biogenesis; specific inhibitors are indicated in red. **k**–**m** LD content in native and acidosis-adapted SiHa cells following treatment with 15 μM A922500 (DGAT1i) or 10 μM PF-06424439 (DGAT2i) for 24 h (**k**), incubation in a medium supplemented with lipid-containing serum (FBS) or with charcoal-delipidated serum (DFBS) in presence or absence of an exogenous lipid concentrate (LC) for 24 h (**l**) or treatment with two distinct CD36 blocking antibodies for 24 h (**m**). Data are represented as mean ± SEM of three independent experiments (with ≥6 technical replicates). Significance was determined by Student's *t*-test (**d**, **e**, **g**), by one-way ANOVA (**k**–**m**) or two-way ANOVA (**i**) with Bonferroni multiple-comparison analysis. ***$p < 0.001$; ns, not significant. Source data are provided as a Source Data file.

an effect mimicked by DGAT1 inhibition and exacerbated when interventions were combined (Supplementary Fig. 2b). Another potential role for LD is to act as energy stores for cancer cells when facing fuel deprivation. We therefore pre-challenged 6.5/cancer cells with the adenylate cyclase activator forskolin to force lipolysis and acutely remove LD from 6.5/cancer cells (Supplementary Fig. 2c). This led us to document that LD deprivation accelerated cell death in 6.5/cancer cells cultured in a low serum-containing medium (Fig. 2b). Instead of removing LD from acidosis-adapted cancer cells, we next inhibited FA release from LD by blocking the activity of adipose triglyceride lipase (ATGL) with atglistatin and found that this treatment similarly accelerated cell death in 6.5/cancer cells cultured in a low serum-containing medium (Fig. 2c and Supplementary Fig. 2d). We next found that the gain in survival of 6.5/cancer cells was lost under hypoxic

conditions (Fig. 2d and Supplementary Fig. 2e), suggesting that oxidation of FA released from LD is needed to support cell survival. Finally, we examined whether LD, by providing an internal source of energy, could help resist to anoikis (i.e., anchorage-dependent cell death). A net effect on the survival of matrix-detached 6.5/cancer cells (i.e., viable cell suspension) was observed vs. 7.4/cancer cells when adhesion was restricted by using low-attachment plate (Fig. 2e, left panel denoted as static) or fully prevented by applying dynamic force (i.e., slight plate shaking during the assay) (Fig. 2e, right panel denoted as dynamic). Strikingly, in these experiments, the presence of exogenous FA in the extracellular medium could not rescue non-anchored 7.4/cancer cells and LD deprivation significantly reduced the observed 6.5/cell survival gain (Fig. 2f). Because anoikis resistance is a major component of the invasive and

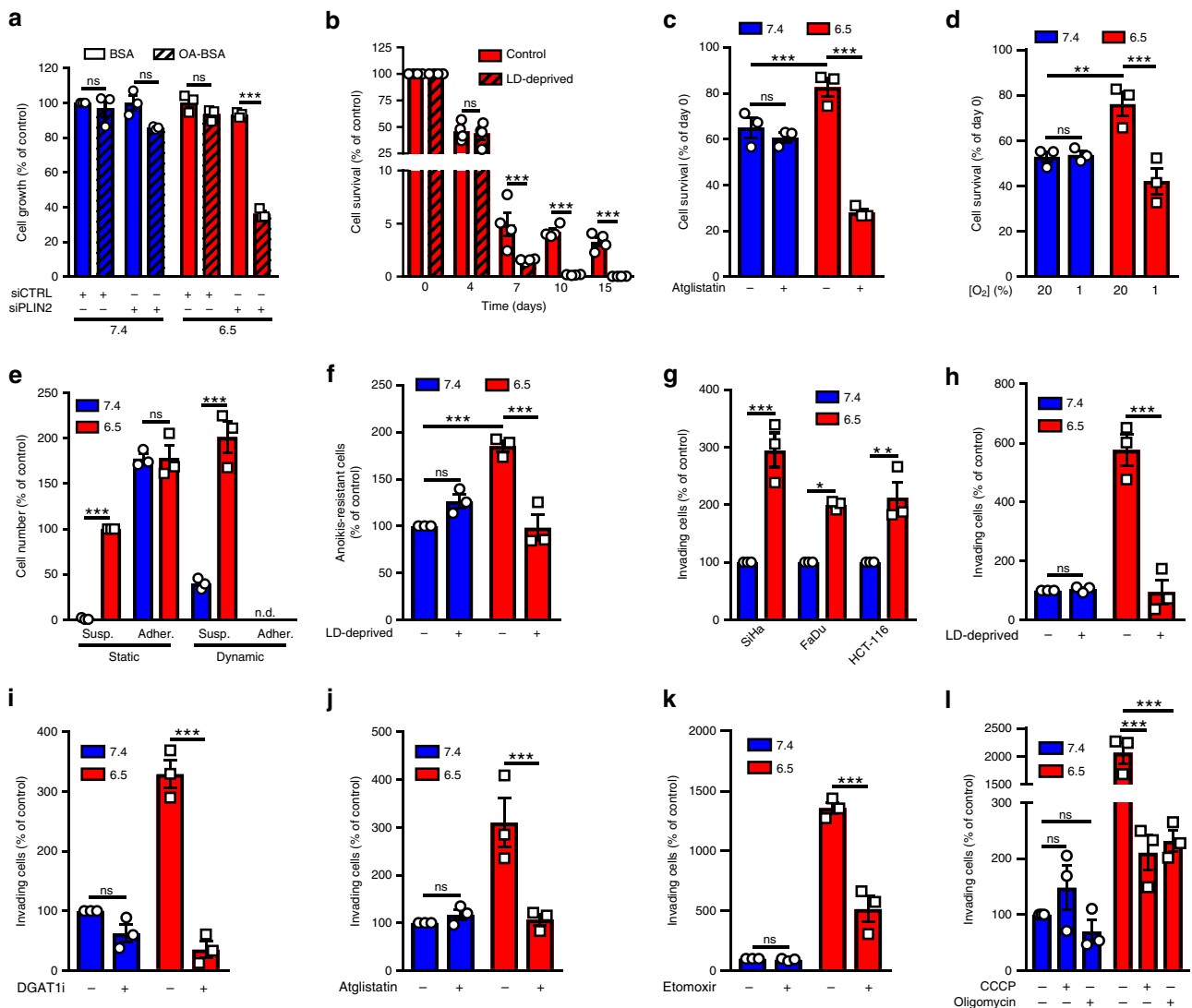

**Fig. 2 Lipid droplets support the survival and invasiveness of acidosis-adapted cancer cells. a** Cell growth extent for native and acidosis-adapted SiHa cells following transfection of PLIN2-targeting (or control) siRNA and treatment with 50 μM BSA-conjugated oleic acid (OA) for 72 h. **b** Survival capacity of LD-deprived acidosis-adapted SiHa cells, at different timings, in low serum-containing medium. **c–e** Survival capacity of native and acidosis-adapted SiHa cells for 3 days in low serum-containing medium following treatment with 10 μM atglistatin (**c**), in a dialyzed serum-containing medium without glucose nor glutamine under normoxic (20% $O_2$) or hypoxic (1% $O_2$) conditions (**d**) or incubation on low-attachment plates, without (static) or with slight shaking (dynamic) (**e**), are depicted the number of cells either adherent or in suspension (i.e., anoikis resistant); n.d., not detectable. **f** Extent of anoikis-resistant cells as determined in Fig. 2e (i.e., cells in suspension) in LD-deprived native and acidosis-adapted SiHa cells. **g** Invasion capacity of native and acidosis-adapted tumor cells in Matrigel-coated Boyden chambers for 24 h. **h–l** Invasion capacity of native and acidosis-adapted SiHa cells in the presence of serum (i.e., an exogenous FA source) upon LD deprivation (obtained by 24 h treatment with 10 μM forskolin) (**h**), 15 μM A922500 (DGAT1i) (**i**), 10 μM atglistatin (**j**), 30 μM etomoxir (**k**) or 100 μM CCCP and 1 μg/ml oligomycin (**l**). Data are represented as mean ± SEM of three independent experiments (with ≥6 technical replicates). Significance was determined by two-way ANOVA (**a–l**) with Bonferroni multiple-comparison analysis. *$p < 0.05$; **$p < 0.01$; ***$p < 0.001$; ns, not significant. Source data are provided as a Source Data file.

metastatic potential of cancer cells, we next examined the invasiveness of acidosis-adapted cancer cells using modified (Matrigel-coated) Boyden chambers. We found that 6.5/cancer cells exhibited a higher invasion potential than corresponding native cancer cells (Fig. 2g). Importantly, this was largely independent of the environmental pH since similar results were obtained by using pH 6.5-adapted cancer cells in a pH 7.4-buffered medium during the time of the invasion assay (Supplementary Fig. 2f), supporting a profound phenotypic alteration (instead of an acute acid-mediated activation of migratory processes). We next addressed the possible roles of LD in the invasive potential of acidosis-adapted cancer cells. We found that upon LD deprivation, either by forskolin or DGAT1 inhibitor treatment (before

the assay), 6.5/SiHa cancer cells exhibited a reduced invasiveness in modified Boyden chambers (Fig. 2h, i); similar results were obtained using HCT-116 cancer cells (Supplementary Fig. 2g, h). Of note, in 6.5/cancer cells, DGAT2 inhibition failed to inhibit invasion (Supplementary Fig. 2i, j). To more directly prove that fatty acid mobilization from LD accounted for the increased invasive potential of 6.5/cancer cells, we inhibited ATGL with atglistatin and documented a similar reduction in invasiveness (Fig. 2j); this experiment was carried out in the presence of serum further confirming that exogenous FA could not rescue the invasive capacity of lipase-inhibited 6.5/cancer cells. Finally, to address the fate of the increased FA release from LD in invading 6.5/cancer cells, we used inhibitors of mitochondrial uptake of

fatty acyl-CoA (etomoxir) (Fig. 2k and Supplementary Fig. 2k) and of ATP generation (oligomycin (ATP synthase inhibitor) and carbonyl cyanide m-chlorophenylhydrazone or CCCP (OXPHOS uncoupler)) (Fig. 2l and Supplementary Fig. 2l). These three different inhibitors dramatically reduced the invasive potential of 6.5/cancer cells. Altogether, these results confirm the direct contribution of LD (and FA mobilization therefrom) to anoikis resistance and increased invasiveness of acid-exposed cancer cells.

**TGF-β2 signaling supports invasiveness and LD formation.** To evaluate how chronic adaptation to acidosis may support the above LD-dependent invasive phenotype, we collected mRNA from 6.5/cell populations for a full transcriptome analysis (vs corresponding 7.4/cells). This led us to identify a list of 349 genes upregulated in each 6.5/cell line while 306 transcripts were significantly downregulated in the same three cell types (FDR < 0.01 and FPKM > 0.5) (Supplementary Fig. 3a and Supplementary Data 1). Interestingly, in the short list of the 20 genes the most upregulated in each of the three screened 6.5/cancer cells, we found *TGFB2* that encodes TGF-β2, which belongs to the family of transforming growth factors often associated with cancer cell invasiveness and the epithelial-mesenchymal transition (EMT)[35], and *FST* (follistatin), a gene under the control of TGF-β2[36] (Fig. 3a). A net increase in TGF-β2 mRNA transcript was confirmed by qPCR in various acidosis-adapted cancer cells (Fig. 3b). ELISA confirmed a significant increase of active TGF-β2 protein levels in acidosis-adapted cancer cell extracts (Fig. 3c) and corresponding extracellular media (Supplementary Fig. 3b). Of note, in the same conditions, active TGF-β1 protein levels remained unchanged (Supplementary Fig. 3c, d). Increases in TGF-β2 mRNA and protein could be recapitulated by 12–24 h exposure of native cancer cells to acidic pH 6.5 (Fig. 3d and Supplementary Fig. 3e) and further evidence of the activation of TGF-β2 signaling pathway in acidosis-adapted cancer cells was obtained by documenting an increase in the extent of phospho-Smad2/3 (Fig. 3e and Supplementary Fig. 3f). Our search for the mechanism driving the preferential increase in TGF-β2 activity under acidosis then led us to document an autocrine TGF-β2-induced TGF-β2 transcription loop (Fig. 3f and Supplementary Fig. 3g) and more importantly, the upregulation of thrombospondin-1 (TSP-1), a critical actor of TGF-β2 maturation (Fig. 3g). Contrary to other TGF-β isoforms that contain a RGD motif in their latency-associated peptide to promote integrin-dependent activation, TGF-β2 activation requires TSP-1 to facilitate maturation in its active form[37]. To further prove a role of TSP-1, we showed that TSP1 silencing reduced the extent of mature (active) TGF-β2 (Fig. 3h) and associated phospho-Smad signaling (Supplementary Fig. 3h).

A role for TGF-β2 in supporting the invasive potential of 6.5/cancer cells was next evaluated using Trabedersen, a specific TGF-β2-targeting antisense oligonucleotide currently evaluated in clinical trials[38–40], and SB431542, a selective TGF-βRI inhibitor. We first documented that Trabedersen reduced TGF-β2 mRNA and protein expression (Supplementary Fig. 3i, j) and that both inhibitors prevented Smad2/3 phosphorylation (Fig. 3i and Supplementary Fig. 3k, l). Importantly, inhibition of TGF-β2 signaling led to a significant reduction in the invasion of acidosis-adapted cancer cells through Matrigel (Fig. 3j and Supplementary Fig. 3m) and prevented LD formation in 6.5/cancer cells (Fig. 3k), an effect paralleled by a reduction in the amounts of neutral lipids (Supplementary Fig. 3n). Of note, addition of recombinant TGF-β2 reversed the effects of Trabedersen on LD formation (Fig. 3l).

**TGF-β2 promotes FA uptake and TG accumulation.** We next focused on the causal link between TGF-β2-induced LD formation and the increased invasive potential of acidosis-adapted cancer cells. To address this question, we first examined how TGF-β2

could drive the formation of LD under acidosis. For this purpose, a lipidomic analysis was carried out to identify potential changes in the FA profile of cancer cells exposed to TGF-β2. We found a net increase in the extent of neutral lipids including saturated and monounsaturated FA in response to TGF-β2 stimulation for 6 h (Fig. 4a, b and Supplementary Table 2) while the pools of phospholipids and free FA were not altered (Fig. 4a). Oil Red O staining confirmed that exposure of native 7.4/cancer cells to TGF-β2 also led to a rapid accumulation of LD (Fig. 4c and Supplementary Fig. 4a). To identify the source of FA involved in LD formation, we next used $^{14}$C-labeled palmitate and showed that the uptake of palmitate by native cancer cells was stimulated upon TGF-β2 treatment (Fig. 4d) and that TGF-β2 silencing in 6.5/cancer cells could prevent palmitate influx, the latter effect being abrogated by the addition of exogenous TGF-β2 (Fig. 4e).

We next aimed to examine whether CD36, DGAT1, and PLIN2 that we had identified as key actors of FA accumulation in LD under acidosis were regulated by TGF-β2. While no change in total CD36 protein expression could be detected by immunoblotting (Supplementary Fig. 4b), using a method based on surface protein biotinylation, we found an increase in the expression of CD36 at the plasma membrane in pH 6.5-adapted cancer cells as well as in native cancer cells exposed to TGF-β2 (Fig. 4f). Blockade of TGF-β2 signaling reduced plasma membrane-associated CD36 expression (Fig. 4g) and using a fluorescent palmitate analog (BODIPY-conjugated $C_{16}$), we confirmed the increased capacity of FA uptake by TGFβ2-stimulated native cancer cells (Supplementary Fig. 4c). Different pathways are known to regulate CD36 translocation, namely AMPK[41], SIRT1[42], and PKC-ζ[43]. Since activation of the two former ones represses TGF-β signaling[44,45], we focused on PKC-ζ which has already been reported to mediate TGF-β signaling[43]. Inhibition of CD36 translocation (in response to either acidic pH or direct TGF-β2 exposure) was observed in the presence of a PKC-ζ pseudo-substrate inhibitor (Fig. 4h, i, j for quantification, Supplementary Fig. 4d–f). Of note, although SIRT1 activity was previously shown to be stimulated under acidosis[14], SIRT1 inhibitor EX-527 failed to show any change in the extent of CD36 translocation (Supplementary Fig. 4g).

We also documented a significant upregulation of DGAT1 mRNA and protein expression in pH 6.5-adapted cancer cells (Fig. 4k, l and Supplementary Fig. 4h) that was completely prevented upon TGF-β2 silencing or TGF-βRI blockade (Fig. 4m and Supplementary Fig. 4i, j); DGAT2 expression was not altered in response to Trabedersen or SB431542 (Supplementary Fig. 4k). Interestingly, the PKC-ζ pseudo-substrate inhibitor (that prevented CD36 translocation) also blocked DGAT1 upregulation in 6.5/cancer cells (Fig. 4n and Supplementary Fig. 4l) and in TGF-β2-treated cancer cells (Supplementary Fig. 4l, m), suggesting that CD36 cell surface expression could be the primary event leading to subsequent upregulation of genes involved in FA handling. To support this hypothesis, we treated 6.5/cancer cells with GW6471, an antagonist of PPARα (a major FA-activated transcription factor regulating FA metabolism) and showed that the expression of DGAT1 but also that of mitochondrial fatty acyl-CoA transporter CPT1 were decreased (Fig. 4o). Furthermore, GW6471 treatment induced the inhibition of 6.5/cancer cell growth but did not alter 7.4/cell proliferation further supporting a major role of FA-induced PPARα activity under acidic conditions (Supplementary Fig. 4n, o). Finally, expression of PLIN2 that we found to be upregulated in acidosis-adapted cancer cells (see Fig. 1h) was inhibited upon TGF-β2 silencing (Fig. 4p). Of note, in a study directly comparing gene expression in healthy and cancerous colorectal tissues, PLIN2 expression (but neither PLIN1, nor PLIN3) strongly correlated with TGF-β2 expression (Fig. 4q); a similar co-expression pattern between TGF-β2 and PLIN2 was also found in human clear cell renal cell carcinoma samples (Supplementary Fig. 4p).

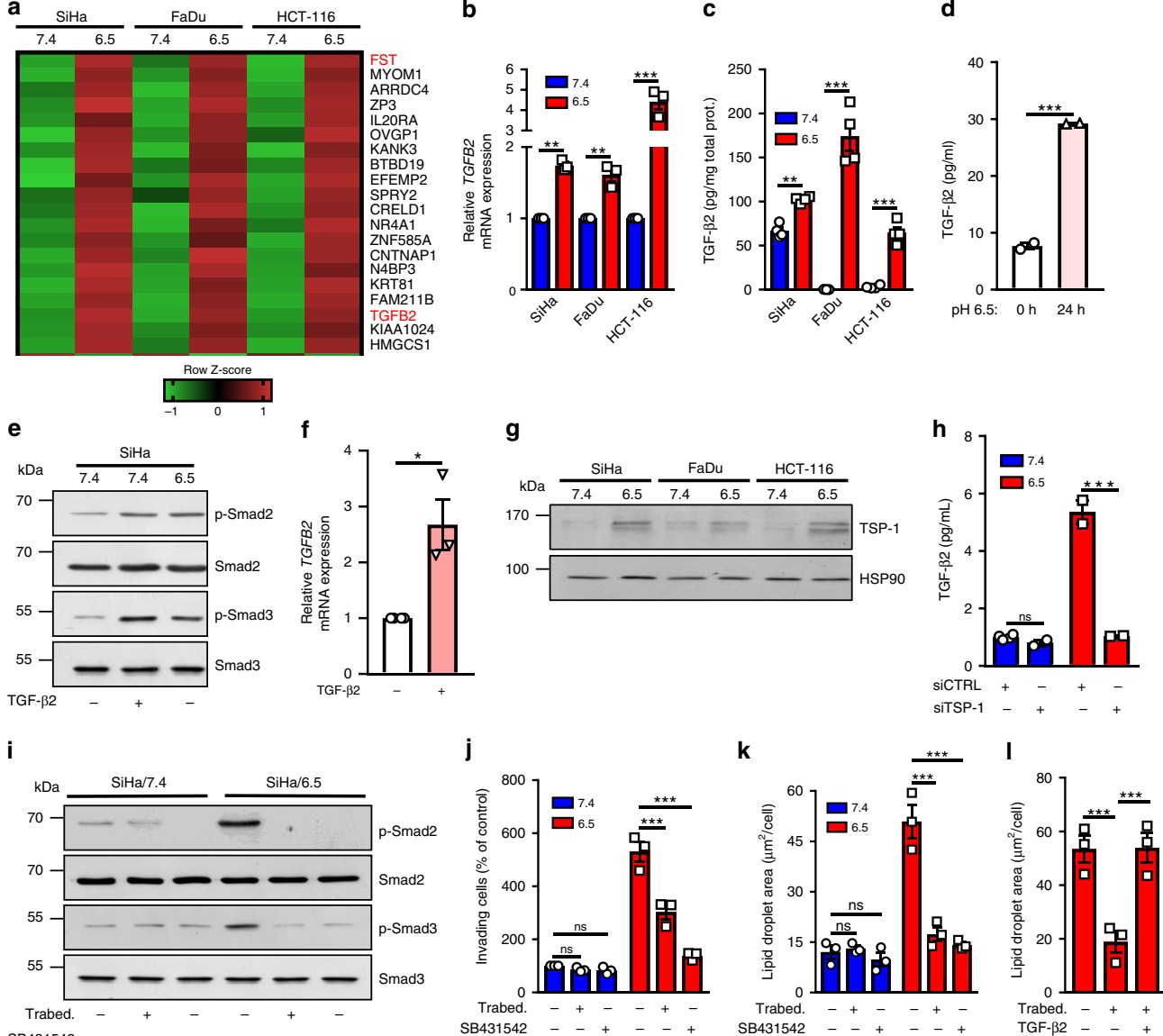

**Fig. 3 TGF-β2 supports invasiveness and LD formation in acidosis-adapted cancer cells. a** Heatmap representation of the top 20 genes upregulated in the indicated acidosis-adapted cancer cells. **b** mRNA expression for *TGFB2* in native and acidosis-adapted tumor cells. **c, d** Levels of active form of TGF-β2 in crude extracts (membrane-bound) from native and acidosis-adapted tumor cells (**c**) or in conditioned media (secreted) from native SiHa cells exposed to acidic pH 6.5 for 24 h (**d**). **e** Representative immunoblotting for phosphorylated and total forms of Smad2 (Ser465/467) and Smad3 (Ser423/425) in native and acidosis-adapted SiHa cells (with or without treatment with 4 ng/ml TGF-β2 for 6 h). **f** mRNA expression for *TGFB2* in native SiHa cells following treatment with 4 ng/ml TGF-β2 for 24 h. **g** Representative immunoblotting for TSP-1 in native and acidosis-adapted tumor cells. **h** Levels of active form of TGF-β2 in conditioned media from native and acidosis-adapted SiHa cells following transfection of TSP1-targeting (or control) siRNA for 72 h. **i–k** Representative immunoblotting for phosphorylated and total forms of Smad3 (Ser423/425) (**i**), invasion capacity in Matrigel-coated Boyden chambers for 24 h (**j**) and LD content (**k**) for native and acidosis-adapted SiHa cells following treatment with 10 μM TGFβ2-specific antisense oligonucleotide Trabedersen for 7 days or 2 μM TGF-βRI inhibitor SB431542 for 24 h. **l** LD content in acidosis-adapted SiHa following treatment with 10 μM TGFβ2-specific antisense oligonucleotide Trabedersen for 7 days in absence or presence of 4 ng/ml TGF-β2 for 24 h. Data are represented as mean ± SEM of three independent experiments (with ≥6 technical replicates). Significance was determined by Student's *t*-test (**d**, **f**), one-way ANOVA (**l**) or two-way ANOVA (**b–h**, **j**, **k**) with Bonferroni multiple-comparison analysis. **p* < 0.05; ***p* < 0.01; ****p* < 0.001; ns, not significant. Source data are provided as a Source Data file.

**EMT is under the control of TGF-β2-stimulated FA metabolism.** We next examined whether LD biogenesis, as driven by TGF-β2, represents a pre-requisite to support invasiveness of acidosis-adapted cancer cells. Since TGF-β2 is known to promote cancer cell invasiveness by inducing EMT, we first aimed to identify molecular markers of this phenotypic transition in 6.5/ cancer cells. We actually showed that in SiHa cancer cells, chronic acidosis inversely influenced the expression of E-cadherin and vimentin, two critical markers of the epithelial and mesenchymal

status, respectively (Fig. 5a). This led us to further explore a variety of EMT markers in different pairs of native and acidosis-adapted cancer cells. Although not all the EMT markers were similarly altered, partial EMT was confirmed in each cancer cell type, with an increase in several mesenchymal traits including N-cadherin (*CDH2*), Snail (*SNAIL1*), Slug (*SNAIL2*), ZEB1 and vimentin, and/or a reduction in ZO-1 and E-cadherin (*CDH1*) as documented by mRNA quantification (Fig. 5b and Supplementary Fig. 5a, b) and protein immunodetection (Fig. 5c and

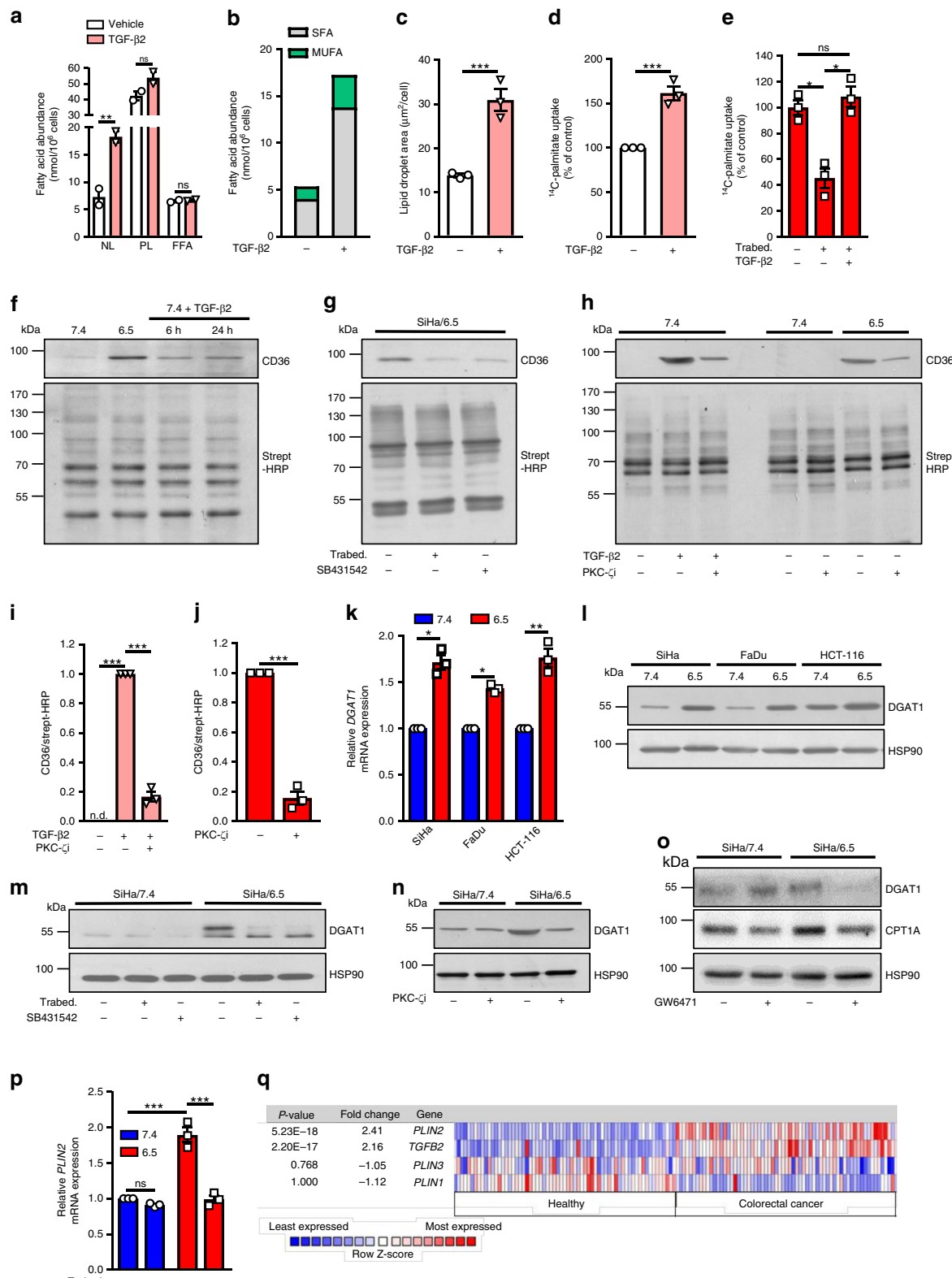

Supplementary Fig. 5c, d). Of note, exposure of 7.4/cancer cells to TGF-β2 recapitulated the EMT signature observed in 6.5/cancer cells (Supplementary Fig. 5e). More importantly, the use of Trabedersen and SB431542 allowed us to confirm that TGF-β2 supports the mesenchymal shift observed in 6.5/cancer cells. Both treatments actually reverted the upregulation of the mesenchymal markers SNAIL1 (Fig. 5d), ZEB1 (Fig. 5e) as well as N-cadherin and vimentin (Fig. 5f and Supplementary Fig. 5f, g) in acidosis-adapted cancer cells. To further prove the role of TGF-β2 in EMT under acidic conditions, we silenced Smad2/3 (Supplementary Fig. 5h, i) and confirmed a dramatic reduction in SNAIL1 and ZEB1 in 6.5/cancer cells (Fig. 5g, h and Supplementary Fig. 5j).

**Fig. 4 TGF-β2 promotes FA uptake and TG accumulation into LD. a–c** Abundance of neutral lipids (NL), phospholipids (PL) and free fatty acids (FFA) (**a**), abundance of saturated and monounsaturated fatty acids (SFA and MUFA, respectively) in the neutral lipid fraction (**b**), and LD content (**c**) in native SiHa cells after treatment with 4 ng/ml TGF-β2 for 6 h. **d, e** [14]C-palmitate uptake for 10 min in SiHa cells after treatment with 4 ng/ml TGF-β2 for 6 h (**d**) and in acidosis-adapted SiHa cells following treatment with 10 μM Trabedersen for 7 days in absence or presence of 4 ng/ml TGF-β2 for 24 h (**e**). **f–h** Representative immunoblotting for cell surface-localized CD36 and total biotinylated proteins in native and acidosis-adapted SiHa cells following treatment with 4 ng/ml TGF-β2 for 6 h and 24 h (**f**), following treatment with 10 μM Trabedersen for 7 days or 2 μM SB431542 for 24 h (**g**) or with 4 ng/ml TGF-β2 and 10 μM PKC-ζ pseudo-substrate inhibitor for 24 h (**h**). **i, j** Quantification of surface-localized CD36 in native and acidosis-adapted SiHa cells treated as indicated in (**h**). **k, l** mRNA (**k**) and protein expression of DGAT1 (**l**) in native and acidosis-adapted tumor cells. **m–o** Representative immunoblotting for DGAT1 in native and acidosis-adapted SiHa cells following treatment with 10 μM Trabedersen for 7 days or 2 μM SB431542 for 24 h (**m**), with 10 μM PKC-ζ pseudo-substrate inhibitor for 24 h (**n**) or with 10 μM GW6471 for 48 h (**o**). **p** mRNA expression of *PLIN2* in native and acidosis-adapted SiHa cells following treatment with 10 μM TGFβ2-specific antisense oligonucleotide Trabedersen for 7 days. **q** Co-expression analysis of *TGFB2, PLIN1, PLIN2,* and *PLIN3* genes in human healthy volunteers and colorectal cancer patient samples. Data are represented as mean ± SEM of three independent experiments (with ≥6 technical replicates). Significance was determined by Student's *t*-test (**c, d, j**), by one-way ANOVA (**e–i**) or two-way ANOVA (**a, k, p**) with Bonferroni multiple-comparison analysis. *$p < 0.05$; **$p < 0.01$; ***$p < 0.001$; ns, not significant. Source data are provided as a Source Data file.

In an attempt to establish a link between EMT and LD accumulation, both being promoted by acidosis-driven TGF-β2 signaling, we could document that silencing ZEB1 blocked EMT in 6.5/cancer cells (Supplementary Fig. 5k) but also reduced the formation of LD under acidosis (Fig. 5i and Supplementary Fig. 5l). We also hypothesized that the other way around, an increase in FA oxidation and associated acetyl-CoA pool could account for Smad2 acetylation, a PTM known to augment its transcriptional activity[46] and to support EMT[47]. Using Seahorse respirometry, we first showed that exogenous FA significantly contributed to oxygen-consumption rate (OCR) in native cancer cells exposed to TGF-β2 as well as in 6.5/cancer cells (vs. 7.4/cancer cells) (Fig. 5j and Supplementary Fig. 5m), an effect largely inhibited upon TGF-β2 silencing or TGF-βRI pharmacological blockade (Fig. 5k and Supplementary Fig. 5n). Stimulated FA oxidation was confirmed by a net increase in the cellular pool of acetyl-CoA in 6.5/cancer cells (Fig. 5l) and associated Smad2 acetylation (Fig. 5m and Supplementary Fig. 5o). Importantly, inhibition of either FA uptake by CD36 blocking antibodies or FA mitochondrial metabolism by CPT1 inhibitor etomoxir significantly reduced Smad2 acetylation in 6.5/cancer cells (Fig. 5n).

Interestingly, when using non-tumorigenic MCF-10A mammary epithelial cells, we failed to document LD accumulation in response to acidosis (Supplementary Fig. 5p) as well as increase in TGF-β2 activity (Supplementary Fig. 5q) and changes in EMT markers (Supplementary Fig. 5r).

**LD supports tumor spheroid growth and metastatic take**. We finally aimed to document that LD accumulation could occur spontaneously in response to ambient acidosis developing in 3D spheroids (i.e., instead of extracellular acidic pH imposed by buffered medium) and could be targeted to reduce metastatic burden in vivo. For 3D tumor spheroid experiments, we used HT-29, a colorectal cancer cell line more prone to form spheroids than the other cell lines used above. We first confirmed that after adaptation to pH 6.5 (Supplementary Fig. 6a), HT-29 cancer cells accumulated neutral lipids within lipid droplets (Supplementary Fig. 6b–f) and exhibited a higher invasiveness potential (Supplementary Fig. 6g) together with an increased TGF-β2 (but not TGF-β1) signaling (Supplementary Fig. 6h–n). We showed that acidic pH spontaneously developed with growth in 3D HT-29 spheroids by using an Alexa568-conjugated pHLIP peptide as a pH sensor (Fig. 6a, b). Interestingly, pHLIP-positive acidic areas significantly overlapped with accumulation of neutral lipids as detected using BODIPY 493/503 staining (Fig. 6c, d) while Ki67 staining revealed that acidic areas were less proliferative than the rim of 3D spheroids (Fig. 6e). To establish a direct link between LD accumulation and TGF-β2 signaling in this 3D model, we exposed spheroids to TGF-βRI blocker SB431542 and

observed a dramatic reduction in BODIPY-stained LD (Fig. 6f, g) together with a growth inhibitory effect (Fig. 6h); the latter was only detectable when spheroids reached a size compatible with spontaneous acidosis development (i.e., between 5 and 10 days; see Fig. 6b, h). Interestingly, we confirmed in sections of primary HT-29 tumors that LD-positive area (stained with ORO) could be identified in acidic tumor regions labeled using Alexa568-conjugated pHLIP peptide (Supplementary Fig. 6o). Of note, although LD could also be identified at the interface between hypoxic and necrotic tumor areas, acidic LD-rich tumor regions did not overlap with tumor hypoxic area (as stained with pimonidazole) (Supplementary Fig. 6o).

We finally examined whether acidic adaptation was associated with a more aggressive phenotype in vivo and importantly whether therapeutic interventions could be derived from the observation of LD generation as part of the EMT phenotype induced by acidosis. For this purpose, we utilized a model of i.v. injection of HT-29 cancer cells to make sure to track the fate of cells whose phenotype resulted from acidosis adaptation and more particularly to evaluate their increased anoikis resistance (see above) when exposed to the bloodstream. We first documented that when compared with corresponding native cells, i.v. injection of acidosis-adapted HT-29 cancer cells led to a more extensive metastatic take in the lungs (Fig. 6I, j). Then, to prove that LD contributed to a higher metastatic burden, we treated mice with etomoxir to prevent FA oxidation to occur in 6.5/HT-29 cancer cells upon i.v. injection. Lung metastatic burden was dramatically reduced in etomoxir-treated mice (Fig. 6k, l). Similar results were obtained with 6.5/FaDu and 6.5/HCT-116 cell lines with a reduced mouse survival following i.v. injection (Fig. 6m, n). With the latter, we could also generate LD-deprived 6.5/cancer cells as detailed in Supplementary Fig. 2d and document that the extent of metastases was significantly reduced (vs. LD-containing 6.5/cancer cells) (Fig. 6o, p).

## Discussion

EMT is a debated concept in oncology mostly because it is not uniformly detected in every metastatic cancer and its contribution to the invasion-metastasis cascade remains influenced and thus confused by the mutation load of primary tumor cells, in particular for the final step of host tissue colonization[48]. Also, contrary to EMT occurring during embryonic development, mesenchymal transition is generally not fully executed in cancer cells. Partial EMT is actually thought to facilitate the necessary reversion to an epithelial phenotype in order to resume proliferation at the metastatic sites[49,50]. This plasticity may account for part of the misperception about the absolute requirement of EMT for cancer progression but also brings an interesting perspective on the role of the microenvironment of primary tumors[51,52]. One may

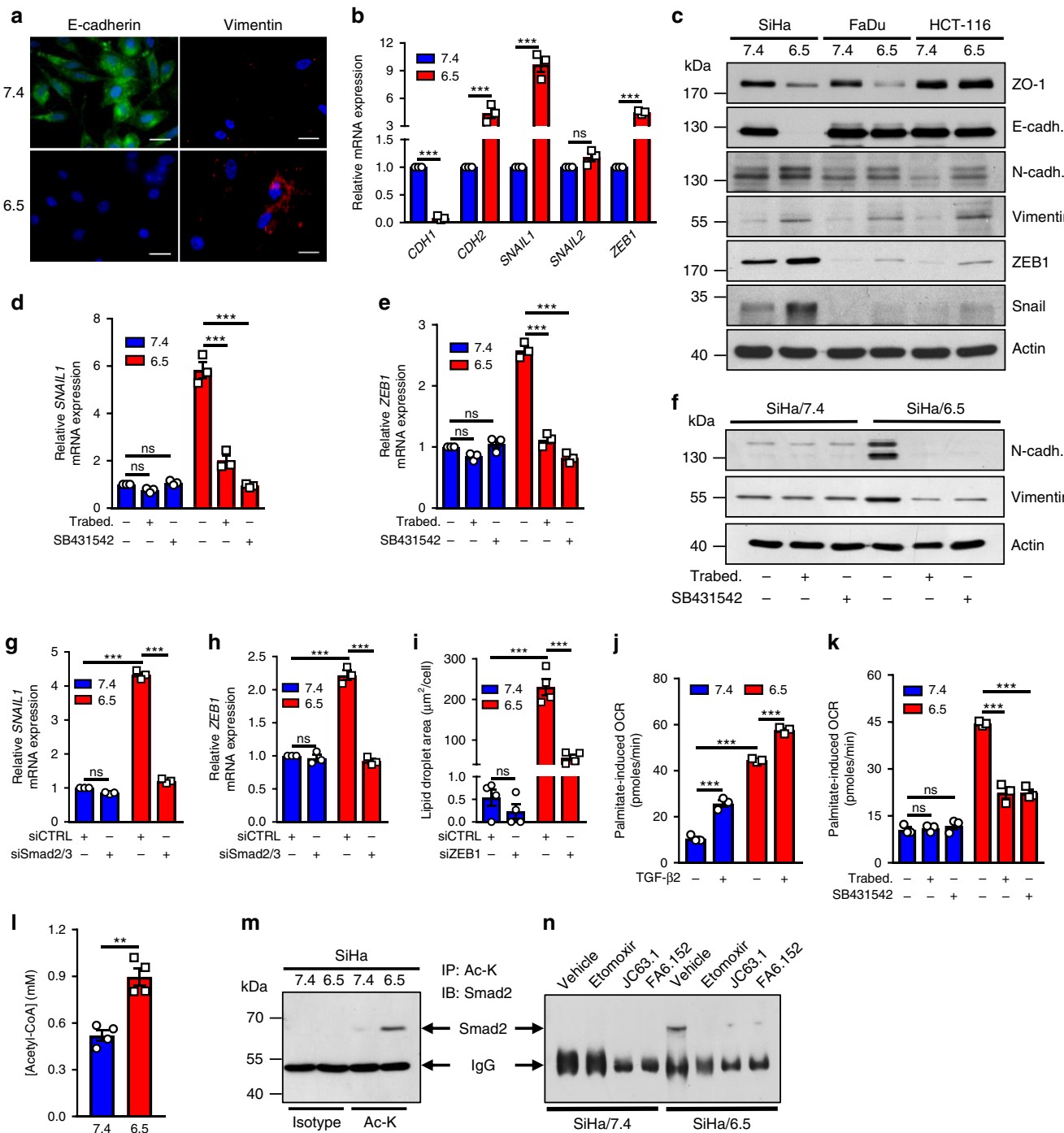

**Fig. 5 EMT in acidosis-adapted cancer cells is driven by TGF-β2. a** Representative immunofluorescence pictures for EMT-related protein markers in native and acidosis-adapted SiHa cells. Scale bar: 20 μm. **b** mRNA expression for epithelial (*CDH1*) and mesenchymal (*CDH2*, *SNAIL1*, *SNAIL2*, and *ZEB1*) genes in native and acidosis-adapted SiHa cells. **c** Representative immunoblotting for some epithelial (ZO-1 and E-cadherin) and mesenchymal (N-cadherin, vimentin, ZEB1, Snail) protein markers in native and acidosis-adapted cancer cell lines. **d, e** mRNA expression for *SNAIL1* (**d**) and *ZEB1* (**e**) in native and acidosis-adapted SiHa cells following treatment with 10 μM Trabedersen for 7 days or 2 μM SB431542 for 24 h. **f** Representative immunoblotting for mesenchymal protein markers in native and acidosis-adapted SiHa cells following treatment as above. **g, h** mRNA expression for *SNAIL1* (**g**) and *ZEB1* (**h**) in native and acidosis-adapted SiHa cells following transfection of Smad2/3-targeting (or control) siRNA for 72 h. **i** LD content in native and acidosis-adapted SiHa cells following transfection with ZEB1-targeting (or control) siRNA for 72 h. **j, k** Palmitate-dependent oxygen-consumption rate (OCR) in native and acidosis-adapted SiHa cells after treatment with 4 ng/ml TGF-β2 for 6 h (**j**) or following treatment with 10 μM Trabedersen for 7 days or 2 μM SB431542 for 24 h (**k**). **l** Levels of cellular acetyl-CoA in native and acidosis-adapted SiHa cells. **m, n** Representative immunoblotting for acetylated Smad2 in native and acidosis-adapted cancer cells (with or without treatment with 4 ng/ml TGF-β2 for 6 h) (**m**) and following treatments with 30 μM etomoxir (CPT1 inhibitor), 2 μg/ml JC63.1 or 1 μg/ml FA6-152 (anti-CD36 blocking antibodies) for 24 h (**n**). Data are represented as mean ± SEM of three independent experiments (with ≥6 technical replicates). Significance was determined by Student's *t*-test (**l**) or two-way ANOVA (**b, d, e, g–k**) with Bonferroni multiple-comparison analysis. **p < 0.01; ***p < 0.001; ns, not significant. Source data are provided as a Source Data file.

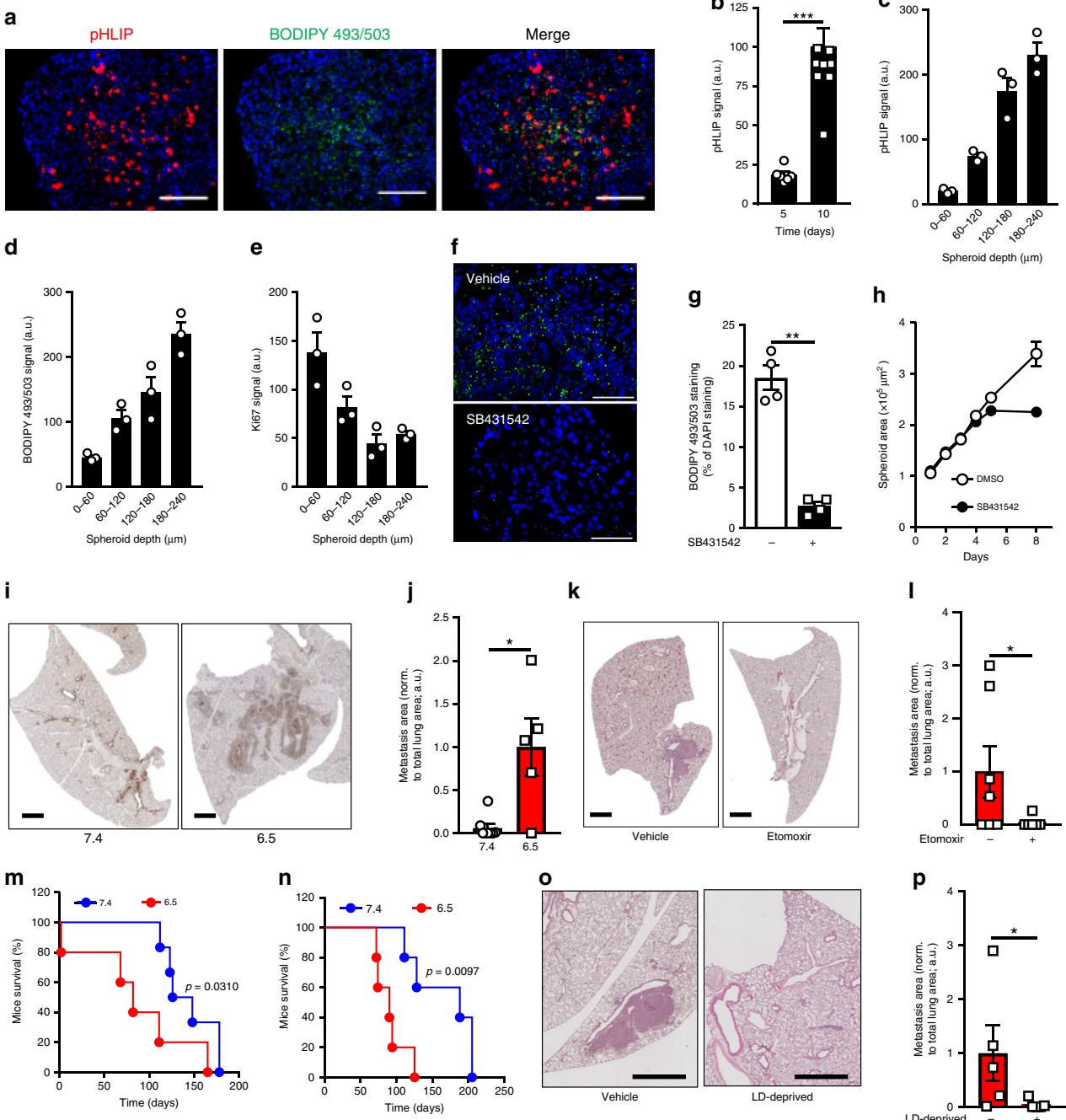

**Fig. 6 TGFβ2-driven LD formation under acidic conditions fosters in vivo cancer cell invasiveness. a** Representative immunofluorescent pictures of Alexa Fluor 568-conjugated pHLIP and BODIPY 493/503 staining in 10-day-old HT-29 spheroids. Scale bar: 100 μm. **b** Time-course accumulation of pHLIP during HT-29 spheroid growth. **c–e** Distribution (from the periphery to the core of the spheroids) of pHLIP (**c**), BODIPY 493/503 (**d**), and Ki67 (**e**) staining. **f, g** Representative immunofluorescent pictures (**f**) and quantification (**g**) of BODIPY 493/503 staining in 10-day-old HT-29 spheroids treated or not with 20 μM TGF-βRI inhibitor SB431542. **h** Time course of the growth inhibitory effects of 20 μM SB431542 on 3D tumor spheroid growth. **i, j** Representative immunohistochemical pictures (**i**) and quantification (**j**) of GFP- or mCherry-positive lung metastases in mice 5 weeks after i.v. injection of GFP-expressing native and mCherry-expressing acidosis-adapted HT-29 cancer cells (n = 5–8 mice per group). Scale bar: 1 mm. **k, l** Representative H&E staining (**k**) and quantification of metastasis area (**l**) in lungs from mice 5 weeks after i.v. injection of acidosis-adapted HT-29 cells and i.p. treatment with 40 mg/kg etomoxir (n = 6–7 mice per group). Scale bar: 1 mm. **m, n** Kaplan–Meier curves depicting mouse survival after i.v. injection of native and acidosis-adapted FaDu (**m**) and HCT-116 (**n**) cancer cells. **o, p** Representative H&E staining (**o**) and quantification of metastasis area (**p**) in lungs from mice 5 weeks after i.v. injection of lipid droplet-deprived (forskolin-treated; or not) acidosis-adapted HCT-116 cells (n = 5 mice per group). Scale bar: 250 μm. Data are represented as mean ± SEM of three independent experiments (with ≥ 6 technical replicates). Significance was determined by Student's t-test (**b, g, j, l, p**). *p < 0.05; **p < 0.01; ***p < 0.001. Source data are provided as a Source Data file.

indeed speculate that TME peculiarities trigger the phenotypic shift required to increase cell motility and initiate invasion but are de facto unable to maintain it when cells have left the primary site, thereby facilitating the reverse transitioning when facing a new physiological distant environment. Here, we identified the formation of lipid droplets in response to tumor acidosis as a major actor of the shift toward a transitory mesenchymal-like, invasive cancer cell phenotype.

Our study pinpoints TGF-β2 as the main driver of the lipid-based metabolic rewiring fostered by tumor acidosis and required to support the energy needs of invading cancer cells. While several studies have identified a role of lipids in EMT[53–56], our study tackles this issue by integrating the TME concept (i.e., tumor acidosis). We previously reported how cancer cells chronically exposed to an acidic pH progressively abandon glycolysis in favor of FAO[15]. Now, we show that this acidosis-guided metabolic preference is dependent on TGF-β2 and further complemented by the formation of LD acting as energy stores that are used to support anoikis resistance and invasiveness. Moreover, we provide evidence that stimulation of FA uptake and oxidation directly supports TGFβ2-induced EMT by increasing the acetyl-CoA pool which in turn promotes the acetylation of Smad2, a PTM associated with an increased activity of this transcription factor[46]. These findings underly a major difference with hypoxia, another TME hallmark also known to promote LD formation[57]. Indeed, the lack of $O_2$ prevents β-oxidation to occur from LD-released FA under hypoxia and tumor reoxygenation is actually required for invasive cancer cells to exploit internal FA stores[57]. The specificity of acidosis is that cancer cells may accumulate FA within LD to reduce lipotoxicity but with the concomitant transition toward the mesenchymal phenotype, they may also directly take advantage of these internal FA stores to generate energy for efficient metastatic spreading.

Expressions of both CD36 and DGAT1, the main entry path for FA in the cytosol and the final actor of their accumulation as neutral lipids, respectively, are regulated by TGF-β2 produced by acidosis-adapted cancer cells. We identified the upregulation of TSP-1 in response to extracellular acidosis as a critical actor of integrin-independent TGF-β2 activation[37]. We further documented a positive feedback loop wherein TGF-β2 transcription is stimulated by TGF-β2 itself, thereby reinforcing TGF-β2 signaling under acidosis. Remarkably, this mode of autocrine, isoform-specific TGF-β2 activation in cancer cells differs from that described in models where a reactive stroma leads to TGF-β secretion[58–60] or where excess free FA uptake itself promotes TGF-β signaling to initiate EMT[56]. In our study, we provide evidence that addition of exogenous TGF-β2 on native cancer cells led to LD formation and that inhibition of TGF-β2 signaling in 3D spheroids undergoing spontaneous acidification dramatically prevented LD accumulation (Fig. 6f–h). Of note, TGF-β1 was reported to shift cell metabolism from FA synthesis to enhanced oxidative phosphorylation[54]. Whether, in the latter study, FA uptake was increased leading to LD formation was, however, not addressed. Although we cannot exclude this possibility, TGF-β2 was in our study the only TGF-β isoform induced in response to ambient acidosis. Thus, collectively, our results allow to propose a model wherein acidosis, by promoting autocrine TSP1-dependent TGF-β2 activation, triggers signaling pathways driving the shift toward a mesenchymal-like invasive phenotype from one hand, and supporting fatty acid uptake, oxidation but also storage in LD from the other hand (Fig. 7). Each arm reinforces each other since FA oxidation favors Smad2 acetylation/activity and thereby supports EMT while the bona fide EMT transcription factor ZEB1 is necessary for LD formation; ZEB-1-driven TGF-β2 expression[61] may actually account for the capacity of TGF-β2 to promote its own gene expression.

Importantly, we documented the druggability of this intrinsic alteration of the cancer cell phenotype with SB431542, a potent and specific inhibitor of TGF-β superfamily type 1 receptor, and AP12009/Trabedersen, a FDA- and EMEA-approved orphan anticancer drug, active as a TGF-β2-specific antisense oligodeoxynucleotide. While TGF-β2 was identified as an attractive target to block oncogenesis resulting from stem cell renewal[62] and encouraging survival results have been reported with Trabedersen in gliomas[38] and pancreatic cancers[39,40], inhibitors of TGF-β2 signaling may actually represent a therapeutic strategy with a broader potential of reducing metastatic burden. Potential therapeutic targets may also be derived from the observation that LD represent a non-dispensable source of fuels to support anoikis resistance. We showed for instance that preventing LD formation in cancer cells facing an acidic pH with a DGAT1 inhibitor represents an achievable goal to limit invasion, offering a rationale for the repurposing of such drugs currently under clinical evaluation for the treatment of type 2 diabetes and obesity[63]. We also found that mouse survival was significantly reduced after injection of acidosis-adapted cancer cells (vs. native cancer cells) and that pre-treatment to remove LD from pH 6.5-adapted cancer cells or acute treatment to prevent the oxidation of FA released from these LD considerably reduced their capacity to form metastases. Together with our data on the supportive role of LD in anoikis resistance and local cancer cell invasion, these findings offer a rationale for the development of drugs inhibiting the mobilization of FA (e.g., atglistatin) to reduce metastatic spreading by preventing the use by cancer cells of the energy stored in preformed LD; the blockade of triglyceride lipase is all the more relevant as it can also interfere with (tumor-induced) release of FA from adipocytes to fuel cancer cell LD[64]. Finally, our observation of a functional CD36 upregulation in response to TGF-β2 is reminiscent of the reported role of CD36 in metastasis-initiating CD44-positive cells in human oral carcinoma[65]. Although CD36-expressing cells in the above study could not be related to EMT, we provide evidence that neutralizing anti-CD36 antibodies could also prevent LD formation in acidosis-adapted cancer cells, making this strategy particularly suited to prevent TGF-β2-driven tumor progression.

In conclusion, the identification of an autocrine TGF-β2-LD axis led us to expand on the driving role of acidosis on metabolic adaptation in cancer cells but importantly to identify new prophylactic and therapeutic perspectives to reduce metastatic burden and thereby extend cancer patient survival.

## Methods

**Cell culture**. Human cervix SiHa (#HTB-35), pharynx FaDu (#HTB-43) and colorectal HCT-116 (#CCL-247) and HT-29 (#HTB-38) cancer cell lines and the normal human breast epithelial cell line MCF-10A (#CRL-10317) were purchased from ATCC. Cells were stored according to the supplier's instructions and used within 6 months after resuscitation of frozen aliquots. All cell lines were maintained in DMEM supplemented with 10% heat-inactivated FBS, 10 mM D-glucose, 2 mM L-glutamine and 25 mM of both PIPES and HEPES before adjusting pH to 7.4 or 6.5. Acidic pH-adapted tumor cells were established as previously described[14,15]. All cell lines were tested for mycoplasma contamination with the MycoAlert™ Mycoplasma Detection kit (#LT07-318; Lonza) before being used. GFP-expressing native and mCherry-expressing acidosis-adapted HT-29 cancer cells were established by infection with eGFP Lentifect™ (#LPP-EGFR-LV105-205; GeneCopoeia) and mCherry Lentifect™ (#LPP-MCHR-LV105-025; GeneCopoeia) purified lentiviral particles according to manufacturer's instructions.

**Cell treatment and transfection**. Cell treatment was performed in a full culture medium with 4 ng/ml recombinant human TGF-β2 (#ab84070; Abcam), 1 μM EX-527 (E7034; Sigma-Aldrich), 10 μM GW6471 (#4618; Tocris), 2 μM SB431542 (#1614; Tocris Bioscience), 10 μM Trabedersen (Eurogentec), 10 μM PKC-ζ pseudo-substrate inhibitor (sc-397537; Santa Cruz Biotechnology), 2 μg/ml JC63.1 (ab23680; Abcam), or 1 μg/ml FA6-152 (ab17044; Abcam) anti-CD36 blocking antibodies at different timings, as indicated in the figure legends. To study lipid droplet biogenesis, cells were first treated with 10 μM forskolin (#F3917; Sigma-Aldrich) for 24 h in a medium supplemented with 10% charcoal-stripped FBS (#F6765; Sigma-Aldrich) to induce lipolysis. After extensive washing, cells were then incubated for 24 h with either a medium supplemented with 10% charcoal-stripped FBS in absence or presence of a chemically-defined lipid concentrate (#11905031; Thermo Fisher Scientific) or with a medium supplemented with 10% lipoprotein-deficient FBS (#S5394; Sigma-Aldrich). A medium supplemented with

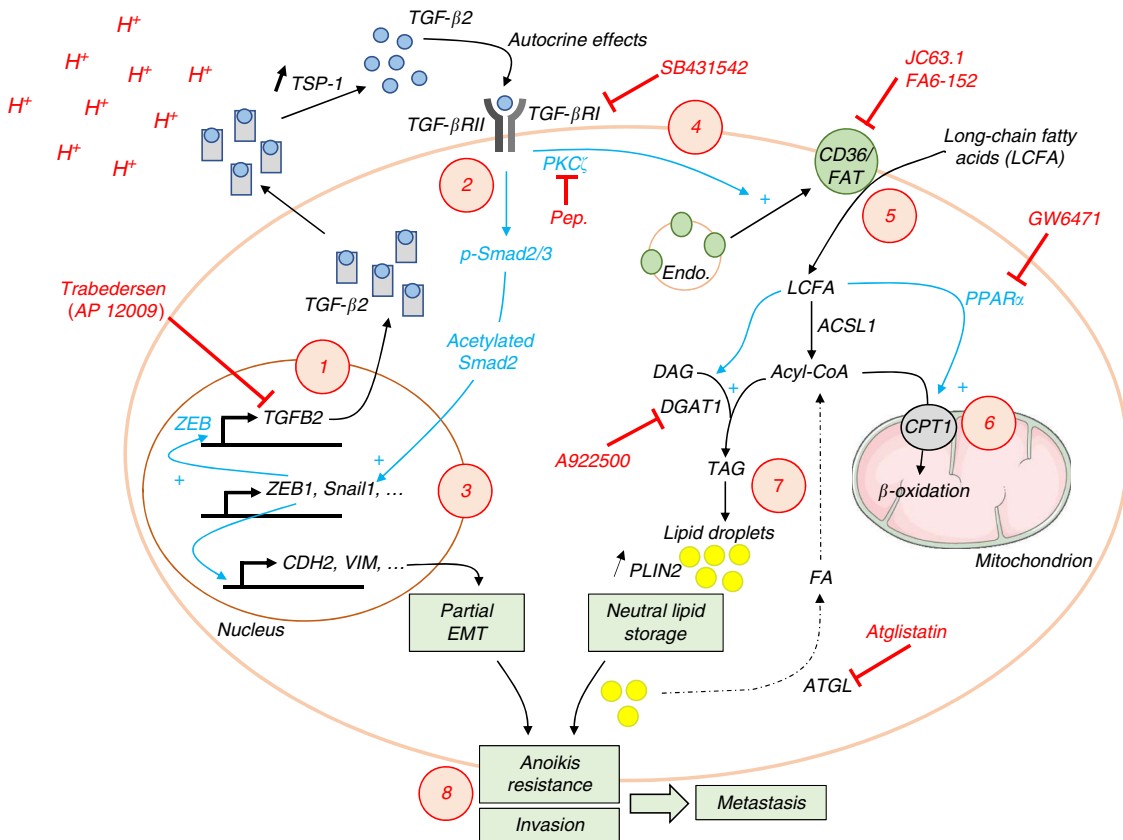

**Fig. 7 Acidosis-induced TGF-β2 supports cancer progression by stimulating both EMT and lipid storage in tumor cells.** Acidosis triggers TGF-β2 upregulation/activation in a TSP1-dependent manner (**1**) and promotes an autocrine signaling pathway through TGF-βRI receptor and subsequent Smad2/3 phosphorylation (**2**) in cancer cells. TGFβ2-mediated signaling pathways induce a partial EMT gene programme (**3**) as well as a PKCζ-dependent translocation of CD36 fatty acid translocase to the plasma membrane (**4**). TGF-β2 autocrine signaling, in acidosis-adapted cancer cells, favors the uptake of long-chain fatty acids (**5**), the latter being oxidized in mitochondria for energy production (**6**) but also stored, as neutral lipids (triacylglycerols) in lipid droplets, in a DGAT1-dependent manner (**7**); the resulting increase in the cellular acetyl-CoA pool accounts for an overall increase in the acetylation of proteins, including Smad2 (thereby increasing its transcriptional activity). Both TGFβ2-induced partial EMT and lipid storage participate to increase survival (i.e., anoikis resistance) and invasion capacities that support in vivo metastatic spreading, ATP needs being fulfilled by FAO upon preferred hydrolysis of TG stored into LD (**8**). Inhibitors able to block different steps of the acidosis-induced TGF-β2/DGAT1 axis are indicated in red. ACSL1: acyl-CoA synthetase long chain family member 1; ATGL: adipose triglyceride lipase; CPT1: carnitine palmitoyltransferase 1; DAG: diacylglycerol; endo: endosomes; FA: fatty acids; PKCζ: protein kinase C ζ; PLIN2: perilipin 2; TAG: triacylglycerol; TSP-1: thrombospondin-1.

10% normal FBS (#F7524; Sigma-Aldrich) was also used for cell treatment with 15 μM A922500 (#A1737), 10 μM atglistatin (#SML1075), 5 μM avasimibe (#PZ0190), 20 μM BPTES, 10 μM PF-06424439 (#PZ0233) and 1 μM simvastatin (#S6196). All inhibitors were from Sigma-Aldrich, except BPTES that was synthesized and purified in our lab[14]. For cell growth assays, oleic acid (#10-1801; Larodan) was conjugated to BSA and added at 50 μM in culture medium for 72 h. In some conditions, cells were maintained at 37 °C in normoxic conditions (20% O$_2$, 5% CO$_2$) or exposed to hypoxia (1% O$_2$, 5% CO$_2$) in an Invivo2 300 hypoxia chamber (Ruskinn). Cell growth was assessed by using the PrestoBlue® cell viability reagent (#A13262; Thermo Fisher Scientific) according to manufacturer's instructions. Cell transfection with a non-targeting pool of 4 siRNA sequences (D-001810-10-05) or a pool of 4 siRNA sequences targeting either human *PLIN2* (#L-019204-01), human *SMAD2* (#L-003561-00), human *SMAD3* (#L-020067-00), human *TSP-1* (*THBS1*) (L-019743-00) or human *ZEB1* gene (L-006564-01), all from Dharmacon, was carried out with Lipofectamine™ RNAiMAX transfection reagent (#13778150; Thermo Fisher Scientific), according to manufacturer's instructions.

**Gene expression analysis in patient samples**. Comparison of expression patterns for *TGFB2*, *PLIN1*, *PLIN2*, and *PLIN3* genes was performed in colorectal and renal datasets using ONCOMINE data-mining platform (https://www.oncomine.org). Sixty-five rectal adenocarcinomas and their paired normal rectal mucosa samples[66] as well as 32 clear cell renal cell carcinoma and 23 normal kidney samples[67] were analyzed for gene expression. Log2 median-centered intensity was used to show the (over)expression of selected genes in tumor samples vs normal samples. Of note, colors used in the chart are z-score normalized to depict relative values within rows. They cannot be used to compare values between rows.

**RNA sequencing data analysis**. Total RNA was extracted using the Maxwell RSC SimplyRNA tissue kit (#AS1340; Promega) according to manufacturer's instructions including a DNAse treatment. RNA quantity and quality were evaluated with a Nanodrop 1000 spectrophotometer (Thermo Fisher Scientific) and RNA 6000 nano chips on a Bioanalyzer 2100 system (Agilent), respectively. RNA sequencing (RNA-Seq) analysis was performed by the VIB Nucleomics Core (Leuven, Belgium). Briefly, samples (biological triplicates) were prepared with the TruSeq RNA Library Prep Kit v2 (#RS-122-2001; Illumina) from 2 μg RNA and sequencing was performed using the NextSeq 500 High Output Kit V2 (75 cycles; Illumina) with single end reads all according to manufacturer's recommendations.

Quality filtering was performed using FastX toolkit 0.0.13 (http://hannonlab.cshl.edu/fastx_toolkit/index.html) and ShortRead 1.16.3. Low quality ends with <Q20 were trimmed and reads shorter than 35 bases after trimming were discarded. Further adapter trimming was performed using Cutadapt 1.7.1. Adapters were trimmed only at the end with at least 10 bases overlap and 90% match. Poly-A reads (more than 90% of the bases equal A), low quality reads (more than 50% of the bases <Q25) and artifact reads (all but 3 bases in the read equal one base type) were all discarded. Reads matching Illumina PhiX sequencing controls were removed following alignment using Bowtie 2.2.4. Sequences were mapped to the human reference genome (GRCh37.73) with TopHat V2.0.13 (https://ccb.jhu.edu/software/tophat/index.shtml) with parameters set; library-type-firststrand, min-intron-length 50, max-intron length 500000, no-coverage-search, no-mixed, read-realign-edit-dist 3. Using SAMtools 1.1, reads that were non-primary mappings or had mapping qualities of 20 or lower were removed from the alignment. Reads overlapping with gene features were identified using FeatureCounts 1.4.6 with parameters set at −Q 0 −s 2 −t exon −g gene_id. Reads that could be attributed to more than one target or not be attributed to any gene were omitted. Gene-segments for which all samples had <1 count per million reads,

were removed. To correct for GC-content, full quantile normalization was applied on bins of GC-content with the EDASeq Bioconductor package. Correction for library size and RNA composition was performed using full quantile normalization with the EDASeq package. For generating the heatmap, relative gene expression Z-scores were subjected to unsupervised clustering using Cluster 3.0 (http://bonsai.hgc.jp/~mdehoon/software/cluster/) by single linkage uncentered correlation clustering of both genes and samples at default settings. The heatmap of obtained cluster data was generated using GraphPad Prism 7 software. Acidosis-regulated genes were classified, according to their cellular function, by KEGG pathway term analysis using DAVID v6.8 software (http://david.abcc.ncifcrf.gov/).

**Oil Red O staining**. Cells grown on coverslips were fixed with 4% (wt/vol) paraformaldehyde (PFA) for 10 min before staining with 3 mg/ml Oil Red O solution in 60% (vol/vol) isopropanol for 20 min. After nuclear counterstain with hematoxylin, bright-field images were acquired at ×63 magnification using an AxioImager.z1-ApoTome1 (Zeiss) and lipid droplet area per cell was evaluated with Fiji software.

**Acetyl-CoA measurements**. Acetyl-CoA concentration in cellular extracts was determined by using the PicoProbe Acetyl-CoA assay kit from Abcam, according to manufacturer's instructions.

**Immunocytochemistry**. Cells grown on coverslips were fixed with 4% PFA for 10 min. After blocking with 5% BSA for 1 h, cells were incubated overnight at 4 °C with anti-E-cadherin and anti-vimentin antibodies (#14472 and #5741, respectively; Cell Signaling Technology). Cells were then incubated with Alexa Fluor 488- and 568-conjugated anti-rabbit secondary antibodies (#A11034 and #A11036, respectively; Thermo Fisher Scientific) for 1 h and nuclei were counterstained with DAPI. For neutral lipid staining, PFA-fixed cells were incubated with 0.5 µg/ml BODIPY 493/503 (#D3922; Thermo Fisher Scientific) for 30 min at room temperature. Slides were prepared with a fluorescence mounting medium (Dako) and staining was visualized with a Zeiss Imager 1.0 Apotome microscope.

**RNA extraction and RT-qPCR**. Samples were collected from an equal number of intact cells in TRI Reagent® (#TR118; Molecular Research Center). RNA was recovered after separation in 1-bromo-3-chloropropane and precipitation with isopropanol, washed with ethanol (70%), resuspended in RNAse-free water, and then quantified by spectrophotometry (Nanodrop 1000, Thermo Fisher Scientific). After reverse transcription on 500 ng of total RNA with the RevertAid Reverse-Transcriptase, oligo-dT and random hexamers (Thermo Fisher Scientific), quantitative PCR amplification was performed on a ViiA™ 7 real-time PCR system (Applied Biosystems) using Takyon Low Rox SYBR® MasterMix dTTP Blue (#UF-LSMT-B0701; Eurogentec). Relative gene expression was calculated using the ddCt method, with GTF2B as reference gene. Gene-specific primers used in this study are listed in Supplementary Table 3.

**ELISA anti-TGFβ1/2**. Active human TGF-β1 and TGF-β2 protein levels were assessed by using dedicated ELISA detection kits (#DB100B and #DB250, respectively; R&D Systems) following the manufacturer's instructions with some modifications. Indeed, samples (conditioned media or cell lysates) were not activated by acid treatment; this modified protocol allowed us to determine the levels of naturally active TGF-β, without the fraction of latent TGF-β being artificially activated to an immunoreactive form with the acid treatment.

**Western blot analysis**. Protein samples were denatured for 5 min at 95 °C with Laemmli sample buffer containing 10% 2-mercaptoethanol. Samples (20 µg per well) were then loaded onto 8–15% acrylamide/bis-acrylamide gels and SDS-PAGE was carried out as previously described[14]. Nitrocellulose membranes were blocked with 5% BSA for 1 h at RT. Primary antibodies against BiP (#3177), CPT1A (#12252), E-cadherin (#3195), N-cadherin (#13116), phospho-Smad2 (#3108); phospho-Smad3 (#9520), phospho-Smad3 (#9520), Smad2 (#5339), Smad3 (#9523), Smad2/3 (#8685), Snail (#3879), vimentin (#5741), ZEB1 (#3396), and ZO-1 (#8193), all from Cell Signaling Technology, were applied at 1:1000 in 5% BSA overnight at 4 °C. Anti-actin (#A5441; Sigma-Aldrich; 1:1000), anti-CD36 (#100011; Cayman Chemical Company; 1:1000), anti-DGAT1 (#ab54037; Abcam; 1:1000), anti-HSP90 (#610419; BD Biosciences; 1:2000), anti-perilipin 1 (#ab3526; Abcam; 1:1000), anti-perilipin 2 (ab78920; Abcam; 1:1000), anti-perilipin 5 (#PA1-46215; Thermo Fisher Scientific; 1:1000), HRP-conjugated streptavidin (#SA10001; Thermo Fisher Scientific; 1:25,000) and anti-TSP1 (ab85762; Abcam; 1:1000) antibodies were also applied in 5% BSA overnight at 4 °C. In some experiments, cell lysates (1 mg) were first pre-cleared with 5 µg rabbit IgG (#02-6102; Thermo Fisher Scientific) and 30 µl Dynabeads Protein G (#10003D; Thermo Fisher Scientific) for 1 h, at 4 °C on an end-to-end roller, and then incubated with 5 µg anti-acetylated lysine (#9441; Cell Signaling Technology) or rabbit IgG (overnight, 4 °C). Immunoprecipitated proteins were pulled down after incubation with 30 µl Dynabeads Protein G (2 h, 4 °C) and then eluted with 60 µl Laemmli 2X.

**Palmitate uptake assays**. Palmitate uptake in native and acidosis-adapted SiHa cells was assessed by two different assays. In the first assay, cancer cells were incubated for 10 min at 37 °C in culture medium containing 50 nM [14C]palmitate (#NEC534050UC; PerkinElmer). Radioactivity was determined in a microplate counter (PerkinElmer Topcount) and raw counts were normalized with the protein content in each well. In the second assay, the culture medium was removed and the cells were washed with PBS before being incubated with 0.2 µM BODIPY™ FL $C_{16}$ (#D3821; Thermo Fisher Scientific) for 15 min at RT. Cells were washed three times with ice-cold PBS and fixed with 4% PFA for 30 min at RT. To determine FA uptake, cells were analyzed by flow cytometry using the FL1-FITC channel (BD FACSCanto™ II; BD Biosciences). Data were analyzed with the FlowJo v10 software.

**Oxygen-consumption rate measurements**. Oxidation of exogenous fatty acids, by native and acidic pH-adapted cells, was assessed by treating the cells with 50 µM BSA-conjugated palmitate in a substrate-limited medium. Oxygen-consumption rate (OCR) was measured using the Seahorse XF96 plate reader (Agilent Technologies). Palmitate-induced OCR was evaluated by calculating the difference of OCR values before and after palmitate addition in each experimental condition.

**3D spheroid models**. Spheroids were prepared and processed as previously described[68]. Briefly, HT-29 cells (1500 cells/well) were seeded in Ultra-Low Attachment 96-well plates (Corning) in DMEM supplemented with 10% heat-inactivated FBS, 10 mM D-glucose and 2 mM L-glutamine. For immunohistochemical studies, spheroids were incubated for 24 h with 2 µM Alexa Fluor 568-conjugated pH-low insertion peptide variant 3 (pHLIP V3; $NH_2$-ACDDQNPWRAYLDLLFPTDTLLLDLLW-COOH[69]) to label acidic regions. Spheroids were then washed twice in PBS, fixed in 4% PFA, harvested and embedded in OCT. Frozen sections (5 µm) were stained with either BODIPY™ 493/503 (#D3922; Thermo Fisher Scientific) or anti-Ki67 antibody (#556003, BD Biosciences). Sections were incubated with Alexa Fluor 568-conjugated anti-mouse secondary antibodies (#A11031; Thermo Fisher Scientific), and nuclei were counterstained with DAPI. Slides were prepared with fluorescence mounting medium (Dako), and staining was visualized with a Zeiss Imager 1.0 Apotome microscope. All spheroid samples from a same experiment were imaged by using the same gain and exposure settings.

**In vivo mouse experiments**. All experiments involving mouse xenograft and experimental metastasis models received the approval of the ethic committee from the Université Catholique de Louvain (approval ID 2016/UCL/MD018) and were carried out according to national care regulations. All experiments were performed with 7-week-old female Rj:NMRI-$Foxn1^{nu/nu}$ mice from Janvier Labs. Before injection, tumor cells (unlabeled, GFP- or mCherry-expressing) were washed twice with PBS, resuspended in 0.9% NaCl ($1 \times 10^6$ cells or $2 \times 10^6$/100 µl for i.v. or s.c. injection, respectively) and then injected into the tail vein or the right flank of the animals. In some experiments, mice were treated i.p. with 40 mg/kg etomoxir (Tocris) for 6 days, including 1 day before cell injection. Long-term mouse survival was then evaluated or the number of metastases was determined 5 weeks post-injection using hematoxylin/eosin (H&E) staining or immunohistochemistry with anti-GFP (#NB600-308; Novus Biologicals) or anti-mCherry (ab167453; Abcam) antibodies on sections from embedded lung samples (9–11 sections were evaluated per lung). Pimonidazole (60 mg/kg) and pHLIP-AF568 (40 µM) were injected i.v. 24 h or 6 h before tumor resection, respectively, and staining on tumor sections (pimonidazole, pHLIP, Oil Red O) was carried out on acetone-fixed samples as previously described[68]. Images were acquired by using a slide scanner (SCN400; Leica) and analyzed with Author 2017.2 software (Visiopharm®).

**Cell-surface protein biotinylation**. Biotinylation of cell-surface proteins was performed as described elsewhere[70] with minor modifications. Briefly, sub-confluent cells were rinsed twice with ice-cold biotinylation buffer (PBS, 1 mM $CaCl_2$, 0.5 mM $MgCl_2$), and then incubated (30 min, 4 °C) with 1 mg/ml EZ-Link™ Sulfo-NHS-SS-biotin (#21331; Thermo Fisher Scientific) to label membrane proteins. After quenching of free biotin with 0.1 M glycine (30 min, 4 °C) two washes with ice-cold biotinylation buffer, cells were lysed for protein extraction. Biotinylated proteins were then isolated by immobilization on a streptavidin agarose resin (#20347; Thermo Fisher Scientific) for 2 h at 4 °C. Beads were washed three times (lysis buffer, 4 °C) and eluted in Laemmli sample buffer containing 10% 2-mercaptoethanol (5 min, 95 °C).

**Invasion assays**. Invasion assays were performed in 12-well Transwell microchambers (Corning) with 8 µm pore-sized membranes, coated with growth factor-reduced Matrigel (#356231; Corning). Cells ($1 \times 10^5$/well for SiHa and FaDu; $2 \times 10^5$/well for HCT-116 and HT-29) were seeded, in the upper chamber of the transwells, in 0.1% FBS-containing medium, in presence or not of the treatments. In some conditions (as described in figure legends), BSA-conjugated oleate and palmitate (25 µM of each; #10-1801 and #10-1600, respectively; Larodan), 15 µM A922500, 10 µM atglistatin or 30 µM etomoxir were added in the upper chamber for the time of the invasion assay (24 h). Otherwise, cancer cells were pre-incubated with 100 µM CCCP or 1 µg/ml oligomycin for 30 min before starting the assay. All inhibitors were purchased from Sigma-Aldrich. A medium containing 10% FBS was added in the lower chamber and the cells were allowed to invade for 24 h at 37 °C. The transwell membranes were then fixed

with ice-cold methanol and stained with DAPI. Cells that had not invaded through the chamber were removed with a cotton swab. Invading cells were imaged by fluorescence microscopy with the Axiovert 100 (CarlZeiss) (excitation filter G365 and emission filter Band Pass 445/50), and four fields were independently counted from each invasion chamber.

**Anoikis resistance assays.** Anoikis resistance was determined by incubating cancer cells on low attachment, hydrophobic 100 mm-plates (#82.1472.001, Sarstedt) either in a static or dynamic mode, the latter being obtained by slight shaking of the plate within the incubator. Cells under suspension after 24 h incubation were collected by centrifugation and adherent cells were detached with trypsin (# 25300-054, Life Technologies) for cell counting (Cellometer, Auto T4 Bioscience).

**Lipidomic analysis.** Lipids were extracted from subconfluent cells following the Folch method with subsequent modifications. Briefly, cells were suspended in PBS and lipids were extracted with chloroform:methanol (1:2, v/v) (VWR Chemicals). Samples were then dried under a stream of nitrogen at 30 °C and resuspended in chloroform. Dosage of cholesterol esters was carried out using the Amplex™ Red Cholesterol Assay kit (#A12216; Thermo Fisher Scientific) according to manufacturer's instructions. Gas chromatography (GC) analysis of fatty acid methyl esters (FAME) was also performed after lipid extraction. Tridecanoic acid, 1,2-dipentadecanoyl-sn-glycero-3-phosphatidylcholin and triheptadecanoin (Larodan) were used as internal standards for free fatty acids, phospholipids and triglycerides quantification, respectively. Solid phase extraction was performed on Bond Elut aminopropyl-modified cartridges to successively retrieve the neutral lipid fraction (triglycerides, diglycerides, monoglycerides and cholesteryl esters), the free fatty acid fraction, and the phospholipid fraction. Extracted fatty acids were converted into FAME via methylation under alkaline conditions (KOH 0.1 M in methanol, at 70 °C for 1 h) and then under acidic conditions (HCl 1.2 M in methanol, at 70 °C for 20 min). FAME were then separated by GC with a GC Trace 1310 (Thermo Fisher Scientific) equipped with a RT2560 capillary colum (100 m × 0.25 mm internal diameter; 0.2 μm film thickness; Restek), an "on-column" automatic injector (CTC Analytics AG) and a flame ionization detector kept at a constant temperature of 255 °C. The system used hydrogen as the carrier gas at an operating pressure of 200 kPa. The GC temperature program was as follows: an initial temperature of 80 °C which progressively increased to 175 °C (for 25 min) at 25 °C/min, then to 200 °C (for 20 min) at 10 °C/min, then to 220 °C (for 5 min) at 10 °C/min, and finally to 235 °C (for 15 min) at 10 °C/min. Each peak was identified by comparison of retention times with those for pure methyl ester standards (Larodan and Nu-Check Prep). Data processing was operated by using the ChromQuest 5.0 software (Thermo Fisher Scientific). Results are expressed in nmol FA per million of cells after data normalization with the background values, the percentage of recovery and the starting cell number.

**Electron microscopy.** For transmission electron microscopy (TEM), subconfluent cells were fixed for 2 h at 4 °C in 2.5% (w/v) glutaraldehyde (Agar Scientific) in 0.1 M cacodylate buffer (pH 7.4). Cells were washed with cacodylate buffer and subsequently post-fixed in 1% (v/w) osmium tetroxide (Merck). Samples were dehydrated by successive passages in increasing concentrated ethanol baths (30, 50, 70, 85, and 100%). After embedding in epon resin LX 112 (Ladd Research Industries), ultra-thin sections of cell-covered filters were prepared using an 8800 ultrotome III (LKB). TEM analysis was then performed at 80 keV (FEI Tecnai 10, Philips) using TEM grids (Agar Scientific) covered with non-porous formvar.

**Statistical analysis.** Statistical analyses were performed using GraphPad Prism 7. Two-tailed unpaired Student $t$-test, one-way or two-way ANOVA tests (Bonferroni's post hoc test) were used where appropriate. For all statistical analyses, the expected variance was similar between the groups that were compared, and significance was accepted at the 95% confidence level ($*p < 0.05$, $**p < 0.01$, $***p < 0.001$).

**Reporting summary.** Further information on research design is available in the Nature Research Reporting Summary linked to this article.

## Data availability

The source data underlying all figures are provided as a Source Data file. The accession number for the RNA sequencing data reported in this paper is GEO: GSE116035. Any further information about resources and reagents should be directed to, and will be fulfilled by the corresponding authors upon reasonable request.

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

## Acknowledgements

This work was supported by grants from the Fonds de la Recherche Scientifique (F.R.S.-FNRS) (PDR T008719F), the Télévie (PDR-TLV 7.8502.18), the Belgian Foundation against cancer (2016-101), the J. Maisin Foundation and an Action de Recherche Concertée (ARC 19/24-096). J.S., E.D., C.V.L., and B.D. are Télévie PhD fellows. C.D. and C.C. are F.R.S.-FNRS Research Associates.

## Author contributions

Conceptualization: C.C., E.B., J.S. and O.F.; methodology: C.C., E.B., J.S., E.D., C.V.L., B.D., C.D. and R.M.; investigation: C.C., E.B., J.S., E.D., C.V.L, B.D., C.Deg., C.G. and L.P.; writing—original draft: C.C. and O.F.; writing—review & editing: C.C. and O.F.; funding acquisition: C.C. and O.F.; resources: C.M., C.D. and Y.L.; supervision: C.C. and O.F.

## Competing interests

The authors declare no competing interests.
