## [Peer Review File · Nature Communications]

Reviewers' comments:

Reviewer #1 (Remarks to the Author):

Metabolic adaptation is a hallmark of cancer. In this manuscript, the authors show the importance of lipid droplets in the invasive potential of cancer cells. They claim that under acidic conditions, notably in acidic adapted cells, cancer cells incorporate fatty acids, which are then stored in the form of triglycerides (lipid droplets). When required, ATGL will mediate the hydrolysis of these triglycerides. Using transcriptomic analysis, they found that TGFB2 was one of the genes which expression was increased upon the adaptation of the cancer cells to the acidic conditions. They prove that lipid droplet formation is dependent on the activity of TGFB2 and they correlate the invasive potential of the cells, with the TGFB2 effects on lipid droplet formation. Finally, using a xenografted mouse model, they show that acidic adapted cells have a higher metastatic potential. The authors used different approaches to prove their hypotheses, and the experimental part is good. The originality of the paper is, however, precluded by previous observations by other labs (Wang et al. 2017 JCI insight, 2017 Feb 23;2(4):e87489; Bensaad et al. 2014 Cell Reports 9, 349-365) showing that breast cancer cells accumulate fatty acids in lipid droplets under hypoxic conditions. Moreover, they show that ATGL plays an essential role in this process, similar to what the authors of the present manuscript found. The mechanisms underlying the effects of fatty acids on the invasion of breast cancer cells would be, however, very original. Unfortunately, this is not addressed enough in this manuscript. In general, I have the following main critics:

1. It is suggested in the manuscript that lipid droplet hydrolysis is required for migration and invasion of the acidic-adapted cancer cells. An important question is how TG hydrolysis and fatty acid oxidation mediate these processes. This would have some relevance.
2. The origin of the incorporated fatty acids is also a remaining question to be addressed. Are other cancer cells providing them? Or are the stromal cancer-associated fibroblasts?
3. The proliferation of cancer cells requires de novo lipid synthesis. This may seem paradoxical with an increase in fatty acid oxidation. Are both processes occurring at the same time in these cancer cells?
4. Similarly, it is nicely shown that these cancer cells require the formation of lipid droplets. But how lipolysis and lipid droplet formation is regulated? Is the same TGFB2 factor that regulates both processes? This is a paradox.

5. In figure 2, the authors analyze the effect of PLIN2 inhibition (siRNA) and lipid droplet deprivation on cell growth and survival. They prove that cancer cells and adapted cells die in response to these treatments. Could this be a consequence of the accumulation of free fatty acids (toxicity) in the cells?

6. When lipolysis is inhibited, adapted cancer cells die. Can this be rescued by glucose or amino acids?

7. In figure 4, it is shown that CD36 protein expression is increased in the acidic-adapted cells, and this enhanced expression is abrogated by the TGFB2 inhibitors. mRNA expression data should also be shown. Moreover, to prove that TGFB2 is directly regulating the expression of these genes (PLIN, DGAT1, CD36, and others), as the authors claim, other methods need to be used, such as promoter analyses, ChIP-PCR, etc.

8. The concept of partial EMT is not clear for this reviewer.

9. There are some inconsistencies in the cell models that are used. Sometimes one cell model is used for one experiment, and other tests are done with other cell models. The authors should do the same experiments in the same cell models.

10. What is the fate of the fatty acids generated upon lipolysis in the cancer cells? Is the formation of Acetyl CoA a critical event? Does the inhibition of ATP synthesis also decrease the invasion potential of the cells?

11. Is there an accumulation of lipid droplets in the tumors grafted in mice?

Reviewer #2 (Remarks to the Author):

This manuscript by Corbet et al., (NCOMM-19-08694) examines the role of acidic pH on the invasive potential of acidosis-adapted cancer cells. The work demonstrates that acidic pH promotes autocrine TGF β 2/Smad signaling promoting formation of lipid droplets (LD) that act as an energy source to support anoikis resistance and the invasive potential of cancer cells. The observations are novel and of potential interest and at the experimental level the work is for the most part well controlled. The delineation and presentation of the work is somewhat scattered and difficult to follow. There are issues and concerns that need to be addressed to support the overall conclusion of the study however; these include:

- Little information regarding how acidic pH initiates the transcriptional induction of TGF β 2 and/or how TGF β 2 becomes activated to initiate the pathway. Both of these points need to be addressed experimentally and/or in the discussion.
- TGF β -mediated EMT has been well described in non-cancerous normal murine mammary (NMuMG) and human mammary (HMLE) cells. Would be of note to establish whether acidosis mediates TGF β 2/Smad signaling in these "normal" cells to mediate LD formation and EMT.
- The TGF β pathway is validated solely through inhibitor studies (Trab. And bIR inhibitor). Smad2/3 silencing (siRNA) should be used for confirmation.
- Is the effect of acidic pH specific for TGF β 2 in multiple cell lines or can TGF β 1 also be induced?
- Does exogenous TGF β 2 induce LD and CD36, DGAT1 & PLIN2 in normal pH media or does this require the acidic pH?

Reviewer #3 (Remarks to the Author):

Nature Communications Review NCOMMS-19-08694

"TGF β 2-induced formation of lipid droplets is necessary for acidosis-driven EMT and metastatic spreading of cancer cells." by Prof Feron

This paper is from a group that has an established reputation in this area of research and have previously shown acidosis effects tumour metabolism and stimulation of fatty acid oxidation. This new work, shows in three cells lines from different types of cancer, acidosis at the extreme end

found clinically, pH 6.5, induces lipid droplet accumulation. They then investigated in depth the mechanisms of formation of lipid droplets, their role in metabolism, invasion and metastasis.

Experiments are well-defined and follow a clear order, where they demonstrate through a series of siRNAs for PLIN2 or inhibitors of many of the key enzymes or proteins involved in uptake of lipids and their degradation, their role in lipid droplet formation and utilisation in the Krebs cycle. This work is clearly summarised in their figure 7, demonstrating the key role of TGF β 2, which can be induced quite rapidly by acid exposure of non-adapted cells and induce a similar phenotype with regard to lipid accumulation. They show induction of EMT by TGF β 2 induces this phenotype and this represents a new step forward in understanding the role of acidosis in tumour metabolism.

However, is EMT necessary and if it is inhibited by targeting, for example, one of the key transcriptional mediators-do lipid droplets still accumulate?

In the first sentence of the results, it might be useful to add an extra sentence to describe how the acidosis adapted cancer cells were developed, without having to go back to the original literature.

Several basic points of clarity are needed:

Firstly, to state at the beginning whether the cell lines that are acid-adapted are always experimented on at pH 6.5 and those that normally grow at 7.4 are always grown at 7.4.

Secondly, none of the western blots are quantified or have statistics applied to them, and I think where key observations are made it is important that we do know the data for three western blots and the p values. Not necessarily for every protein on every gel, but in principle, showing the difference between 7.4 and 6.5.

Perhaps the most key point that needs to be understood is the induction of TGF β 2 by acid pH. This appears acutely in pH 7.4 cell lines. As this could occur after 12 or 24 hours, it would be interesting to know whether such a short time frame could also induce the changes in invasion that were reported in the chronic adaptation. They do make the point that the pH 6.5 adapted cells at pH 7.4, will still invade more than 7.4 parent cells, supporting a persistent phenotype, but it is important to clarify the pH in the other experiments, maybe at the beginning of the results section.

Line 119, particularly resistant to rather than reluctant to.

Perhaps the most difficult part of the complete story here is how acid generates increased TGF β 2. For example, the RNA goes up much less than the protein, could it be that there are stores in the extracellular matrix that are released under acid conditions or that the processing of TGF β 2 precursors is stimulated by acid.

Line 295, we utilised a model rather than privileged a model.

Figure on survival-which cell lines were used?

The spheroid model, acid pH in the centre associated with lipid droplet accumulation, but the centre is also hypoxic so what could the contribution of hypoxia be versus acidosis here?

We should know what the gene lists are for the analyses showing supplementary figure 3 A.

The in vivo experiments were convincing in that pre-treating the acid-adapted cells before injecting them reduced metastasis. However, much more important is whether the acid induced cells, which have increased metastasis, can be reverted by blocking lipid metabolism. I think this is a key experiment to show the potential utility of this approach is to use the acid-adapted cells and show that there is a reduced tumour burden with any one of the drugs or antibodies that they previously used.

Reviewer#1:

Metabolic adaptation is a hallmark of cancer. In this manuscript, the authors show the importance of lipid droplets in the invasive potential of cancer cells. They claim that under acidic conditions, notably in acidic adapted cells, cancer cells incorporate fatty acids, which are then stored in the form of triglycerides (lipid droplets). When required, ATGL will mediate the hydrolysis of these triglycerides. Using transcriptomic analysis, they found that TGB2 was one of the genes which expression was increased upon the adaptation of the cancer cells to the acidic conditions. They prove that lipid droplet formation is dependent on the activity of TGFB2 and they correlate the invasive potential of the cells, with the TGFB2 effects on lipid droplet formation. Finally, using a xenografted mouse model, they show that acidic adapted cells have a higher metastatic potential. The authors used different approaches to prove their hypotheses, and the experimental part is good. The originality of the paper is, however, precluded by previous observations by other labs (Wang et al. 2017 JCI insight, 2017 Feb 23;2(4):e87489; Bensaad et al. 2014 Cell Reports 9, 349-365) showing that breast cancer cells accumulate fatty acids in lipid droplets under hypoxic conditions. Moreover, they show that ATGL plays an essential role in this process, similar to what the authors of the present manuscript found. The mechanisms underlying the effects of fatty acids on the invasion of breast cancer cells would be, however, very original. Unfortunately, this is not addressed enough in this manuscript.

We are pleased that the Reviewer appreciates our experimental work and we thank him/her for the constructive comments. We believe that new findings obtained in response to the Reviewer's comments considerably strengthen our manuscript.

As a preamble, we would like to address the general comments about to the related work by others that the Reviewer identified. While our study is about the influence of tumor acidosis, the work by Bensaad is focused on hypoxia. It should first be stressed that the tumor acidic compartment does not completely overlap with the hypoxic one (see Corbet & Feron, Nature Rev Cancer 2017); we now also provide an illustration of this dichotomy in new Suppl. Fig.6o. Second, FA oxidation requires O₂ and may thus occur under acidic conditions as long as there is enough O₂ available while oxidative metabolism is hampered under hypoxia. In the work of Bensaad et al., reoxygenation is actually needed to observe FAO as fueled by FA released from lipid droplets. To illustrate this difference, we now provide evidence that when facing nutrient deprivation, the survival of LD-containing cells is increased under normoxia while it is reduced under hypoxia where the lack of O₂ prevents beta-oxidation to occur (new Fig. 2d and new Suppl. Fig. 2e). It should also be mentioned that LD formation under hypoxia is driven by HIF-1 α in the work by Bensaad et al. whereas upon acid exposure, HIF-1 α activity is strongly inhibited as we previously reported (Corbet et al. Cancer Res 2014) accounting for a strong decrease in glycolytic flux and an increase in FA metabolism under acidosis. We took advantage of the Reviewer's comment to include hypoxia in the discussion of our revised manuscript.

About the paper by Wang, these authors reported that adipocytes provide FA to cancer cells that are stored before hydrolysis but in their model, the increase in FAO is associated with a decreased ETC activity (ie, uncoupled FAO with lesser ATP produced and accumulation of Krebs cycle intermediates) and an increased glycolysis/lactate release from glucose. They conclude that this eliminates the role of an increase in energy supply to support invasion and

refer to potential epigenetic alterations to explain the increased invasiveness. These findings are different from ours where ATP generated from OXPHOS is needed for invasion. As requested by the Reviewer (see below), we now provide evidence that in our hands, blocking ATP production prevents invasion of acid-adapted cancer cells (new Fig. 2k-l and new Suppl. Fig. 2k-l). Despite these discrepancies between the work of Wang and ours, we now emphasize in our conclusion that triglyceride lipase (eg, ATGL) represents a potential therapeutic target to impede cancer progression by preventing the release of FA either directly from cancer cells (our work) or from adipocytes (Wang's paper).

In general, I have the following main critics:

1. It is suggested in the manuscript that lipid droplet hydrolysis is required for migration and invasion of the acidic-adapted cancer cells. An important question is how TG hydrolysis and fatty acid oxidation mediate these processes. This would have some relevance.

We agree with the Reviewer that although our initial work suggested that TG hydrolysis and FAO are needed to provide ATP to support invasiveness of acid-adapted cancer cells, clear evidence were lacking. We now provide a new set of experiments documenting the inhibition of 6.5/cell invasion upon blockade of either ATP generation with oligomycin (ATP synthesis inhibitor) and CCCP (uncoupler of respiration from ATP synthesis) (new Fig. 2l and new Suppl. Fig. 2l), as well as blockade of FAO with etomoxir (new Fig. 2k and new Suppl. Fig. 2k).

2. The origin of the incorporated fatty acids is also a remaining question to be addressed. Are other cancer cells providing them? Or are the stromal cancer-associated fibroblasts?

As suggested by the Reviewer, differences in the source of FA according to cancer types certainly exist because of specific tissue environments (eg, adipocytes for breast cancer, CAF for stroma-rich cancers). In the revised version of our manuscript, we now report that (i) FA accumulation into LD is directly proportional to the presence of free FA in the extracellular medium (new Fig. 1l and new Suppl. Fig. 1n) and (ii) lipoprotein-deprived serum did not prevent LD formation (new Suppl. Fig. 1l). These data indicate that LD are mostly generated from free FA whatever the source of them. One may thus propose that the more free FA are available (because of the vicinity with a FA source), the more likely will be the formation of LD in proximal cancer cells. We now quote the work of Wang et al. documenting that FA released from adipocytes after lipolysis, can be transferred and stored in cancer cell as triglycerides.

3. The proliferation of cancer cells requires de novo lipid synthesis. This may seem paradoxical with an increase in fatty acid oxidation. Are both processes occurring at the same time in these cancer cells?

We have recently reported in a paper published in Cell Metabolism (Corbet et al, 2016) that under acidosis, histone deacetylation leads to the downregulation of ACC2, thereby removing the brake of malonyl-CoA (generated along FA synthesis (FAS)) on CPT1, and in turn allowing the concomitance of FAS and FAO in the same cell. We further showed in this

article that under acidosis, glutamine (instead of glucose) was the major driver of FAS (through reductive carboxylation of α -KG). Reference to this work is now clearly mentioned in the introduction of our revised manuscript.

4. Similarly, it is nicely shown that these cancer cells require the formation of lipid droplets. But how lipolysis and lipid droplet formation is regulated? Is the same TGFB2 factor that regulates both processes? This is a paradox.

Our data support a model wherein TGF-beta2 promotes the uptake of FA (Figs 4d-e) that are used for FAO (Figs 5j-k) while the large excess of captured FA is stored into LD (see for instance Figs 3k and 4c). In parallel, TGF-beta2 induces partial EMT (see Fig. 5), thereby increasing the invasive *potential* of acid-adapted cancer cells. It is only when LD-loaded cells face specific ATP needs to survive under nutrient deprivation or to actively support invasion, that lipolysis from LD is induced (see inhibitory effects of atglistatin in Fig. 2c and 2j).

Of note, concomitant TG synthesis and hydrolysis has been described in both normal and pathological conditions. For example, in cardiac myocytes, the majority of lipids oxidized by mitochondria are first esterified into TG, then hydrolyzed before oxidation (Banke et al., *Circ Res.* 2010;107(2):233–241). In this last study, overexpression of PPAR α was shown to increase TG turnover, as well the expression of enzymes involved in TG synthesis (e.g., gpam, agpat, and dgat1). In cancer cells also, Nomura et al. (*Cell.* 2010;140(1):49–61.) have shown that newly synthesized FFAs are immediately converted into neutral lipid stores and that the use of FFAs is dependent on their release.

Also, we realize that confusion may have arisen from the use of DGAT inhibition to prevent the accumulation of LD before using these pre-challenged cells in the invasion assay (Fig. 2i) [and not during the invasion assay]. This is now better explained in the revised version of our manuscript.

5. In figure 2, the authors analyze the effect of PLIN2 inhibition (siRNA) and lipid droplet deprivation on cell growth and survival. They prove that cancer cells and adapted cells die in response to these treatments. Could this be a consequence of the accumulation of free fatty acids (toxicity) in the cells?

In our original manuscript, we have indeed documented that oleic acid becomes toxic for cells incapable of handling excess neutral lipids (Fig. 2a). To further support this hypothesis, we have now documented an increase in ER stress (as detected by BiP expression) in 6.5/cancer cells exposed to oleic acid or to an inhibitor of DGAT1, an effect further exacerbated when the two interventions are combined (new Fig. S2b).

6. When lipolysis is inhibited, adapted cancer cells die. Can this be rescued by glucose or amino acids?

When the release of FA from lipid droplets is blocked, neither glucose, nor glutamine can rescue anoikis. Yet more striking, the presence of exogenous FA in the extracellular medium cannot rescue the capacity of 6.5/cancer cells to invade neither, pointing out the major role of internal stores of FA for survival (Fig. 2e)

7. In figure 4, it is shown that CD36 protein expression is increased in the acidic-adapted cells, and this enhanced expression is abrogated by the TGFB2 inhibitors. mRNA expression data should also be shown. Moreover, to prove that TGFB2 is directly regulating the expression of these genes (PLIN, DGAT1, CD36, and others), as the authors claim, other methods need to be used, such as promoter analyses, ChiP-PCR, etc.

We now provide evidence that total CD36 *expression* is not altered under acidosis (Suppl. Fig. 4b) indicating that only *translocation* and not upregulation of CD36 is induced by acidic pH (Fig. 4f-h and Suppl. Fig. 4d). Since this translocation is a major trigger of the increased FA uptake under acidosis, we further explored the possible mechanism driving CD36 trafficking. Among the different actors known to support CD36 translocation, we considered AMPK, SIRT1 and PKC- ζ . Since activation of the two former ones represses TGF-beta signaling, we focused on PKC- ζ that has already been reported to mediate TGF-beta signaling (Gunaratne et al., Mol Cell Biol 2013). Inhibition of CD36 translocation (in response to either acidic pH or direct TGF-beta2 exposure) was observed in the presence of PKC- ζ pseudo-substrate inhibitor (new Figs. 4h-j and new Suppl. Fig. 4d-f). Of note, although SIRT1 activity was previously shown to be stimulated under acidosis (Corbet et al., Cancer Res 2014), SIRT1 inhibitor EX527 failed to show any change in the extent of CD36 translocation (new Suppl. Fig. 4g).

Interestingly, the PKC- ζ pseudo-substrate inhibitor (that prevented CD36 translocation) also blocked DGAT1 upregulation in 6.5/cancer cells (new Fig. 4n and new Suppl. Fig. 4m) and in TGF-beta2-treated cancer cells (new Suppl. Figs. 4l-m), suggesting that CD36 cell surface expression could be the primary event leading to subsequent upregulation of genes involved in FA handling. To support this hypothesis, we treated 6.5/cancer cells with GW6471, an antagonist of PPARalpha (a major FA-activated transcription factor regulating FA metabolism) and showed that the expression of DGAT1 but also that of mitochondrial fatty acyl-CoA transporter CPT1 were decreased (new Fig. 4o). Furthermore, GW6471 treatment induced the inhibition of 6.5/cancer cell growth but did not alter 7.4/cell proliferation further supporting a major role of FA-induced PPARalpha activity under acidic conditions (new Suppl. Fig. 4n-o).

8. The concept of partial EMT is not clear for this reviewer.

Most tumor cells do not undergo a full EMT, but rather adopt some traits of mesenchymal cells and maintain some epithelial characteristics. Induction of a partial-EMT favors cancer cell invasion and spreading in distant organs while maintaining the ability to reverse the EMT process (the so-called MET), thereby generating more epithelial progeny, whose presence greatly increases the success of metastatic colony formation. Partial EMT has therefore to be understood as a mode of increased plasticity for cancer cells. We have now better explained this concept, including references to recent reviews (Aiello and Kang, J Exp Med 2019; Chaffer (Weinberg) et al., Cancer Metastasis Rev 2016).

9. There are some inconsistencies in the cell models that are used. Sometimes one cell model is used for one experiment, and other tests are done with other cell models. The authors should do the same experiments in the same cell models.

In the revised version of the manuscript, we have uniformized the cell models used throughout our study. Data related to HT-29 cells used in 3D spheroids and most *in vivo* studies are now concentrated in new Suppl. Fig. 6; our aim is here to show that the most salient results obtained *in vitro* with the three other cell lines used along the rest of our study, were also confirmed in this more tumorigenic and more spheroid-prone cell line.

10. What is the fate of the fatty acids generated upon lipolysis in the cancer cells? Is the formation of Acetyl CoA a critical event? Does the inhibition of ATP synthesis also decrease the invasion potential of the cells?

As detailed above, we now provide evidence that TG hydrolysis and FAO are used to provide ATP to acid-adapted cancer cells using inhibitor of mitochondrial acyl-CoA uptake (ie, CPT1 inhibitor etomoxir) (new Fig. 2k and new Suppl. Fig. 2k) and blockers of ATP synthesis (oligomycin and CCCP) (new Fig. 2l and new Suppl. Fig. 2l).

11. Is there an accumulation of lipid droplets in the tumors grafted in mice?

Lipid droplets can be identified in most tumor xenografts. To further support our model that lipid droplets are not only the consequences of hypoxia, we used a pHLP peptide as marker of tumor acidic regions together with pimonidazole as a marker of hypoxia. In Suppl. Fig. 6o, we selected a tumor section in which ORO-positive lipid droplets can be identified in a pHLP-positive acidic tumor area that does not overlap with hypoxia-positive regions as well as in a pimonidazole-positive area.

Reviewer#2:

This manuscript by Corbet et al., (NCOMM-19-08694) examines the role of acidic pH on the invasive potential of acidosis-adapted cancer cells. The work demonstrates that acidic pH promotes autocrine TGF β 2/Smad signaling promoting formation of lipid droplets (LD) that act as an energy source to support anoikis resistance and the invasive potential of cancer cells. The observations are novel and of potential interest and at the experimental level the work is for the most part well controlled. The delineation and presentation of the work is somewhat scattered and difficult to follow.

We are pleased that the Reviewer found our work novel and of interest. In the revised version of our manuscript, we have addressed the different comments and organized the new sets of data to render the manuscript easier to follow.

There are issues and concerns that need to be addressed to support the overall conclusion of the study however; these include:

- Little information regarding how acidic pH initiates the transcriptional induction of TGF β 2 and/or how TGF β 2 becomes activated to initiate the pathway. Both of these points need to be addressed experimentally and/or in the discussion.

The mode of TGF- β 2 activation is different from that of the other TGF- β forms. Because of the lack of RGD in its latency-associated peptide, activation is integrin-independent and a role of thrombospondin has been reported to account for TGF- β 2 maturation. In the revised manuscript, we provide evidence that thrombospondin 1 (TSP-1) is upregulated in acid-adapted cancer cells (new Fig. 3g). Also, silencing TSP-1 is associated with a reduction in the generation of mature TGF- β 2 (new Fig. 3h) and an inhibition of downstream signaling cascade as revealed by the reduction in phospho-Smad (new Suppl. Fig. 3h). In addition, we provide new evidence that TGF- β 2 *per se* promotes the expression of TGF- β 2 in acidic conditions (new Fig. 3f). These data support a model wherein TSP1-driven TGF- β 2 maturation together with a positive transcriptional feedback loop (ie, TGF- β 2-induced TGF- β 2) account for the observed efficient TGF- β 2 signaling under acidosis.

- TGF β -mediated EMT has been well described in non-cancerous normal murine mammary (NMG) and human mammary (HMLE) cells. Would be of note to establish whether acidosis mediates TGF β 2/Smad signaling in these "normal" cells to mediate LD formation and EMT.

To address this question, we used non-tumorigenic MCF-10A mammary epithelial cells, a widely used *in vitro* model for studying normal breast cell function and EMT. We found that acidic pH failed to promote LD generation (new Suppl. Fig. 5p), TGF- β 2 upregulation (new Suppl. Fig. 5q) and EMT (new Suppl. Fig. 5r), indicating that the phenotype we have observed in cancer cells of various origins is cancer-specific.

- The TGF β pathway is validated solely through inhibitor studies (Trab. and bIR inhibitor). Smad2/3 silencing (siRNA) should be used for confirmation.

To abide by the Reviewer's comment, we have now used Smad2/3 siRNA and confirmed that the upregulation of Snail1 and Zeb1 in 6.5/cancer cells was completely abrogated upon Smad silencing (new Figs 5g-h and new Suppl. Figs. 5h-j).

- Is the effect of acidic pH specific for TGF β 2 in multiple cell lines or can TGF β 1 also be induced?

We have observed in four different cancer cell lines, a similar upregulation of TGF-beta2 under acidosis (Figs. 3b-c and Suppl. Fig. 6h-i) and a lack of TGF-beta1 induction (Suppl. Figs 3c-d and Suppl. Fig. 6k-l). As mentioned above, evidence of thrombospondin-1 upregulation further supports the preferred mode of activation of TGF-beta2.

- Does exogenous TGF β 2 induce LD and CD36, DGAT1 & PLIN2 in normal pH media or does this require the acidic pH?

Several figures in the revised manuscript document the capacity of exogenous TGF-beta2 to promote LD formation (Figs 4b-c) and to induce FA uptake (Figs. 4d-e), CD36 translocation (Fig. 4f) and DGAT expression (Suppl. Fig. 4l) at a physiological pH (ie, independently of acidic conditions).

Reviewer #3 (Remarks to the Author):

This paper is from a group that has an established reputation in this area of research and have previously shown acidosis effects tumour metabolism and stimulation of fatty acid oxidation. This new work, shows in three cells lines from different types of cancer, acidosis at the extreme end found clinically, pH 6.5, induces lipid droplet accumulation. They then investigated in depth the mechanisms of formation of lipid droplets, their role in metabolism, invasion and metastasis.

Experiments are well-defined and follow a clear order, where they demonstrate through a series of siRNAs for PLIN2 or inhibitors of many of the key enzymes or proteins involved in uptake of lipids and their degradation, their role in lipid droplet formation and utilisation in the Krebs cycle. This work is clearly summarised in their figure 7, demonstrating the key role of TGF β 2, which can a be induced quite rapidly by acid exposure of non-adapted cells and induce a similar phenotype with regard to lipid accumulation.

We thank the reviewer for the nice comments. In the revised version of the manuscript, we have addressed the different comments.

They show induction of EMT by TGF β 2 induces this phenotype and this represents a new step forward in understanding the role of acidosis in tumour metabolism. However, is EMT necessary and if it is inhibited by targeting, for example, one of the key transcriptional mediators-do lipid droplets still accumulate?

We agree that this piece of evidence was lacking in our original manuscript. In the revised version of our manuscript, we have thus silenced ZEB1 and documented that this led to a dramatic reduction in the extent of lipid droplets in 6.5/cancer cells (new Fig. 5i and new Suppl. Fig 5l) together with the expected EMT inhibition (as revealed by vimentin downmodulation and E-cadherin re-expression) (new Suppl. Fig. 5k). This finding reinforces the interplay between LD and EMT, both under the control of TGF- β 2. Collectively, our results allow to propose a model wherein acidosis, by promoting autocrine TGF- β 2 signaling, triggers signaling pathways driving the shift towards a mesenchymal-like invasive phenotype from one hand, and supporting fatty acid uptake, oxidation but also storage in LD from the other hand (Figure 7). Each arm reinforces each other since metabolism of FA stored in LD leads to Smad2 acetylation and thereby supports EMT while the *bona fide* EMT transcription factor ZEB1 is necessary for LD formation; ZEB-1-driven TGF- β 2 expression (Gregory et al. Mol Biol Cell 2011)) may actually account for the capacity of TGF- β 2 to promote its own gene expression as reported in our study.

In the first sentence of the results, it might be useful to add an extra sentence to describe how the acidosis adapted cancer cells were developed, without having to go back to the original literature.

The text has been edited (first paragraph of the Results section) to include this comment.

Several basic points of clarity are needed:

Firstly, to state at the beginning whether the cell lines that are acid-adapted are always experimented on at pH 6.5 and those that normally grow at 7.4 are always grown at 7.4.

The text has been edited to include this comment (first paragraph of the Results section). Of note, in one set of experiments only, acid-adapted cancer cells were used in an invasion assay at neutral pH to make the point that the cell phenotype (metabolism) of acid-adapted cancer cells and not specific acid-driven processes (eg, pH-dependent proteases) were involved in the increased migratory potential of acid-adapted cancer cells.

Secondly, none of the Western blots are quantified or have statistics applied to them, and I think where key observations are made it is important that we do know the data for three western blots and the p values. Not necessarily for every protein on every gel, but in principle, showing the difference between 7.4 and 6.5.

Western blot quantification has been performed and provided mostly as Supplementary data (new Suppl. Fig 1i, Suppl. Fig 3f, 3k-l, Suppl. Fig 4e-f, h-l, Suppl. Fig 5c-d, f-g and Fig. 4i-j) to avoid making the text more cumbersome.

Perhaps the most key point that needs to be understood is the induction of TGF β 2 by acid pH. This appears acutely in pH 7.4 cell lines. As this could occur after 12 or 24 hours, it would be interesting to know whether such a short time frame could also induce the changes in invasion that were reported in the chronic adaptation.

Our search for the mechanism driving the preferential increase in TGF-beta2 activity under acidosis led us to document an autocrine TGF-beta2-induced TGF-b2 transcription loop (Fig. 3f and Suppl. Fig. 3g) and more importantly, the upregulation of thrombospondin-1 (TSP-1), a critical actor of TGF-beta2 maturation (Fig. 3g). Because of the lack of RGD in its latency-associated peptide, activation is integrin-independent and a role of thrombospondin has been reported to account for TGF-beta2 maturation. In the revised manuscript, we provide evidence that thrombospondin 1 (TSP-1) is upregulated in acid-adapted cancer cells (new Fig. 3g). Also, silencing TSP1 is associated with a reduction in the generation of mature TGF-beta2 (new Fig. 3h) and an inhibition of downstream signaling cascade as revealed by the reduction in phospho-Smad (new Suppl. Fig. 3h). In addition, we provide new evidence that TGF-beta2 *per se* promotes the expression of TGF-beta2 in acidic conditions (new Fig. 3f). These data support a model wherein TSP1-driven TGF-beta-2 maturation together with a positive transcriptional feedback loop (ie, TGF-beta2-induced TGF-beta2 expression) account for the observed efficient TGF-beta-2 signaling under acidosis.

As indicated by the Reviewer, the activation of TGF-beta2 under acidic conditions is rapid and a trend towards an increased invasiveness is observed for 7.4/cancer cells acutely exposed to pH 6.5 (Suppl. Fig. 2f). It is however difficult in these conditions to exclude well known acidic pH-dependent protease activation that supports cancer migration. This is the reason why in the same figure, we have documented the maintenance of the invasive potential of 6.5/cancer cells when the assay is performed at physiological pH, further supporting that profound, persistent metabolic alterations actually drive the invasive phenotype.

They do make the point that the pH 6.5 adapted cells at pH 7.4, will still invade more than 7.4 parent cells, supporting a persistent phenotype, but it is important to clarify the pH in the other experiments, maybe at the beginning of the results section.

As stated above, this information is now clearly provided in the revised manuscript.

Line 119, particularly resistant to rather than reluctant to.

Text edited accordingly.

Perhaps the most difficult part of the complete story here is how acid generates increased TGF β 2. For example, the RNA goes up much less than the protein, could it be that there are stores in the extracellular matrix that are released under acid conditions or that the processing of TGF β 2 precursors is stimulated by acid.

See response related to thrombospondin and TGF- β 2-induced TGF- β 2 expression above.

Line 295, we utilised a model rather than privileged a model.

Text edited accordingly.

Figure on survival-which cell lines were used?

Text edited accordingly. Cell survival and anoikis resistance were evaluated using SiHa and HCT116 cell lines.

The spheroid model, acid pH in the centre associated with lipid droplet accumulation, but the centre is also hypoxic so what could the contribution of hypoxia be versus acidosis here?

This is a well taken comment. Hypoxic cells are actually also described to accumulate lipid droplets (Bensaad et al., Cell Reports 2014). However, it should be stressed that FA oxidation requires O₂ and may thus occur under acidic conditions as long as there is enough O₂ available while oxidative metabolism is hampered under hypoxia. To illustrate this difference, we now provide evidence that when facing nutrient deprivation, the survival of LD-containing cells is increased under normoxia while it is reduced under hypoxia where the lack of O₂ prevents beta-oxidation to occur (new Fig. 2d and new Suppl. Fig. 2e). In the work of Bensaad et al., reoxygenation is actually needed to observe FAO as fueled by FA released from LD.

It should also be mentioned that LD formation under hypoxia is usually driven by HIF-1 α whereas upon acid exposure, HIF-1 α activity is strongly inhibited as we previously reported (Corbet et al. Cancer Res 2014), accounting for a strong decrease in glycolytic flux and an increase in FA metabolism under acidosis. More generally, the tumor acidic compartment does not completely overlap with the hypoxic one (see Corbet & Feron, Nature Rev Cancer 2017); we now provide an illustration of this dichotomy in new Suppl. Fig.6o.

We should know what the gene lists are for the analyses showing supplementary figure 3 A.

Gene lists are now provided in a new Supplementary Table 4.

The in vivo experiments were convincing in that pre-treating the acid-adapted cells before injecting them reduced metastasis. However, much more important is whether the acid induced cells, which have increased metastasis, can be reverted by blocking lipid metabolism. I think this is a key experiment to show the potential utility of this approach is to use the acid-adapted cells and show that there is a reduced tumour burden with any one of the drugs or antibodies that they previously used.

To abide by the Reviewer's comment, we have now used etomoxir to treat mice at the time of i.v. injection of acid-adapted cancer cells. This led to significant reduction in the extent of metastases (new Figs. 6k-l). Together with the failure of LD-deprived acid-adapted cancer cells to form metastases (Figs. 6o-p), these data further document the critical role of LD as an energy reservoir for spreading cancer cells and metastatic progression.

REVIEWERS' COMMENTS:

Reviewer #1 (Remarks to the Author):

The authors have properly addressed all the comments of this reviewer. The manuscript provides new important information that highly contributes to the understanding of the metabolic changes in cancer development and progression. I do not have further major critics.

Reviewer #2 (Remarks to the Author):

The authors have addressed my previous concerns and issues to satisfaction.

Reviewer #3 (Remarks to the Author):

They have very thoroughly responded to all my comments and I think to the other reviewers. In this process they discovered further novel mechanisms and pathways, which enhance the paper. However with regard to the gene lists, as such they are of no value, I had expected them to be ranked by fold change and FDR significance.

Reviewer#1

The authors have properly addressed all the comments of this reviewer. The manuscript provides new important information that highly contributes to the understanding of the metabolic changes in cancer development and progression.

Thank you. No further comment.

Reviewer#2

The authors have addressed my previous concerns and issues to satisfaction.

Thank you. No further comment.

Reviewer#3.

They have very thoroughly responded to all my comments and I think to the other reviewers. In this process they discovered further novel mechanisms and pathways, which enhance the paper. However with regard to the gene lists, as such they are of no value, I had expected them to be ranked by fold change and FDR significance.

We thank the Reviewer for the nice comments. In the final version, we now present the list of genes ranked by fold change together with FDR information. Full data from the RNA sequencing analysis are also archived in GEO under the accession number GSE116035.